# Optimized Design of Earth Dams: Analysis of Zoning and Heterogeneous Material in Its Core

**José Sánchez-Martín [1], Rubén Galindo [1,*], Carlos Arévalo [1], Ignacio Menéndez-Pidal [1], Liliya Kazanskaya [2] and Olga Smirnova [3]**

[1] Departament of Geotechnical Engineering, Escuela Técnica Superior de Caminos, Canales y Puertos, Universidad Politécnica de Madrid, 28040 Madrid, Spain; jsanchezmartin@alumnos.upm.es (J.S.-M.); carlos.arevalo@upm.es (C.A.); ignacio.menendezpidal@upm.es (I.M.-P.)

[2] Russia Department of Building materials and technologies, Emperor Alexander I Petersburg State Transport University, 190031 Saint-Petersburg, Russia; yalifa@inbox.ru

[3] Department of Constructing Mining Enterprises and Underground Structures, Saint-Petersburg Mining University, 199106 Saint-Petersburg, Russia; smirnovaolgam@rambler.ru

*  Correspondence: rubenangel.galindo@upm.es

**Abstract:** To control the seepage in the design of an earth dam, guidelines prescribe a high proportion of fines and high homogeneity of geotechnical characteristics in the material used for the dam core. However, on many occasions there is no material of this nature near the dam placement and, from an economic or environmental point of view, it is not possible to locate and transport material with good geotechnical characteristics close to the dam. This research demonstrated the possibility of using impermeable materials in earth dam cores, as well as soils considered unsuitable according to the classic recommendations and guidelines. For an optimized design, two situations are analyzed here. First, we examined the possibility of using soil with a marked difference in grain size as the core of the dam, each with homogeneous geotechnical properties. In this case, the optimal zoning of up to three types of materials was studied to ensure adequate seepage control. Second, we examined the use of soil with great geotechnical heterogeneity, which presents high permeability dispersion. In such a case, the conditions that would allow its use were studied via the of Montecarlo analysis. By maintaining the soil's global heterogeneity, it was possible to study an unlimited disposition of layers of different permeability. In the first situation, the results showed that the most effective zoning for decreasing seepage flow corresponded with three vertically set materials. In this design, the most optimized zoning (minimal seepage flow rates) corresponded to the most impermeable soil situated downstream when water heights were under 90% of the height of the dam core. However, for maximum water height, more optimized cases corresponded to the intermediate permeability material located downstream. In the second situation, when heterogeneous materials were used to construct the impervious element of the dams, the Montecarlo analysis indicated that the seepage flow rates were limited to sufficiently low values despite the large dispersion of material permeability. In addition, the highest maximum hydraulic gradients were observed in the lowest lifts of the dam core and for situations in which the seepage flow rates were moderate and low.

**Keywords:** heterogeneous material; earth dam; seepage; Darcy's law; optimization

## 1. Introduction

The construction of earth dams requires prior identification and availability of materials in areas close to dam sites, including a zoning study of the materials used for different parts of the dam, e.g., core, upstream/downstream shells, filters, drains, transitions, protections, etc.



By having a zoned dam body, earth dams can allow for a better use of nearby available materials. The most resistant, heavy, and permeable materials are used in the construction of upstream and downstream shells, whereas the most impervious materials are used for the core and the most granular materials are used to constitute the layers of filters, drains, and transitions. Despite this, on many occasions there are insufficient volumes of low-permeability materials in the dam site to cover the construction needs of the dam core. In these cases, resorting to distant quarries to transport low-permeability materials would be economically impractical and would also generate an environmentally unsustainable construction effort. Thus, a zoning of the dam core could be strategically carried out by placing materials with low available permeabilities to optimize dam safety [1].

Frequently, in other situations, the material available at the dam site does not have the optimal and desirable geotechnical characteristics for regular core material, and it is presented as a heterogeneous material. This material shows highly variable properties at its source and its characterization presents significant uncertainties. These materials have a very extensive granulometry, with the presence of particles of very different sizes, with permeabilities that vary over a wide range. In these situations, it is doubtful that these heterogeneous materials could be used safely for the construction of the dam core.

The positive and negative socioeconomic impacts of dam construction were analyzed by Aladelokun [2]. It is a well-known fact that dam construction impacts are multidisciplinary—e.g., climatology, geology, biodiversity, etc.—with each having feedback effects. Environmental managers and planners must, therefore, seek to mitigate negative impacts of dam construction with the use of cognate environmental management plans. In particular, a key aspect is the use of materials in the vicinity of the dam location. Since all soils are pervious to a smaller or larger extent, however, many earth-filled dams are vulnerable to internal erosion and piping due to seepage problems that take place in the core.

Internal erosion is a major cause of failure in embankment dams; specifically, it is the second common cause after overtopping [3]. This phenomenon is strongly influenced by the behavior of clays, which are used as seepage control material because of its low permeability [4–6]. Some types of clay, by their physicochemical properties, have unstable behavior in contact with water. Clay particles separate from each other, thus becoming suspended in the aqueous medium. The main problem is that once in suspension, cracks can propagate, facilitating filtration and consequently failures of piping, concentrated leaks, etc.

As a safety measure, filters are usually available. Granular filters are required to perform two basic functions in embankment dams: prevent the migration of base soil particles and allow drainage of seepage water. Traditionally, retention function is evaluated by using particle size distribution and drainage function is evaluated by using permeability. Only a few authors [7] have used filter permeability for the assessment of retention function and there is no general agreement among them, but permeability should be an important variable because it takes into account other important characteristics such as compaction, porosity, density, and particle shape.

Although the study of seepage through earth-fill dam needs an onsite investigation of hydrological and geological conditions, numerous studies have been conducted using physical models [6,8], because physical models give a general picture of seepage behavior through earth-fill dams, including the phreatic line and the flow rate. However, as physical modeling has many limitations and constraints, numerical modeling, which is based on the mathematical solutions, is the other way used in many studies [9–11] to solve the most complex engineering problems, including seepage research.

One of the ways to control the seepage problem in earth dams is by using proper materials for the core section since the core section of earth dams provides an impermeable barrier within the body of the dam. Thus, Kanchana and Prasanna [12] analyzed the use of various materials with different combinations to zone-type earth dams with a central impervious vertical core, and to study the behavior of the phreatic line at the downstream phase by varying the effective length of the horizontal drainage filter. Al-Janabi et al. [13] investigated the seepage through earth-filled dams using physical, mathematical, and numerical models. The results from the three methods

revealed that both mathematical calculations using analytical solutions and the numerical model resulted in a plotted seepage line compatible with the observed seepage line in the physical model. Seepage analysis revealed that if a sufficient quantity of silty sand soil is available around the proposed dam location, a homogenous earth-fill dam with a medium drain length of 0.5 m thickness is a good design configuration.

Khassaf [14] argued that seepage can cause weakening in the earth dam structure, followed by a sudden failure due to piping or sloughing, and presented a study to determine the quantity of seepage, exit gradient, hydraulic gradient, and pressure head of zoned earth dams under the effect of changing core permeability and core thickness. Zahedi and Aghajani [15] investigated the seepage behavior through the body of an earth-filled dam with different core shapes using a 2D finite element seepage analysis. For this end, a model of an earth-filled dam with different heights and three core types of vertical, inclined, and diaphragm was considered. Within this same type of investigation, Majeed [16] presented a study to simulate the seepage flow through a zoned earth-fill dam. Hassan Al [17] presented an application of finite element analysis to predict the two-dimensional steady-state water seepage through an earth dam with two soil zones resting on an impervious base. Choi [18] presented an evaluation of seepage quantity of an earth dam using 3D finite element analysis. In addition, Jamel [19] presented an investigation of the amount of seepage through an homogenous earth dam core by finite element software. Continuing with this type of study, Fattah et al. [20] made a seepage analysis of a zoned earth dam by finite elements, studying the effect of several parameters, including the permeability of the embankment materials and the presence of an impervious core, together with its location and thickness.

Hellström et al. [21] presented a study of fluid mechanics of internal erosion in embankment dams. This study was divided into two parts. Part one dealt with homogenous materials, which included a general background to porous media flow, continuum models for flow through embankment dams, and methods to numerically model the flow. Part two included models to obtain forces on individual particles, starting from dilute (single particle) systems and ending with very dense and periodic systems of particles.

The present paper provides an analysis to evaluate the seepage behavior of cored earth-filled dams. The first approach describes construction with different types of materials, zoning the core to control the seepage flows according to the height of the core, the inclination of the upstream and downstream faces of the core, the value of the coefficient of permeability of the materials, and the height of water in the reservoir. The second approach examines the use of heterogeneous material in the context of seepage flow through the core and the gradients developed as a consequence of the flow of water, which indicates the behavior of the core against undesirable internal erosion problems that could occur.

The main advance of this research is to demonstrate that it is possible to use soils that are considered unsuitable according to classic recommendations and guidelines as an impervious material in earth dam cores. In this way, for locations where the ideal impermeable material is scarce, it can be economized in some cases and permit the construction of dams in others by using the material from the dam site or nearby areas.

## 2. Analysis Methodology

### 2.1. Theoretical Bases

The phenomenon of water seepage through a soil was established, under certain conditions, by the Laplace equation [22]. To undertake the study of seepage flow through the dam core, the following starting hypotheses were considered to be valid:

- Plain deformation. The two-dimensional problem with water flow in the directions defined within the dam cross section was considered (Figure 1). Therefore, the seepage flow per linear meter of the dam was studied and the total flow defined by the total length of the dam constructed in the

dam site. In Figure 1, the water level is represented at the top of the dam core, without freeboard. As indicated later in Section 3.2, different calculations were made to account for different situations of the reservoir and to be able to see the sensitivity of the filtration flow results with the increase in water heights.

- The filtration calculation was considered decoupled with the stress-strain analysis. Therefore, its compactness did not vary throughout the filtration process.
- Darcy's law was considered valid [23] and, therefore, the speed of the water flow in each point of the soil was proportional to the permeability of the soil and to the variation of the hydraulic potential with respect to its trajectory. It should be pointed out that due to the dam's own construction process, which was carried out by compacted layers of 20 to 40 cm thick. The permeability was considered not be isotropic.

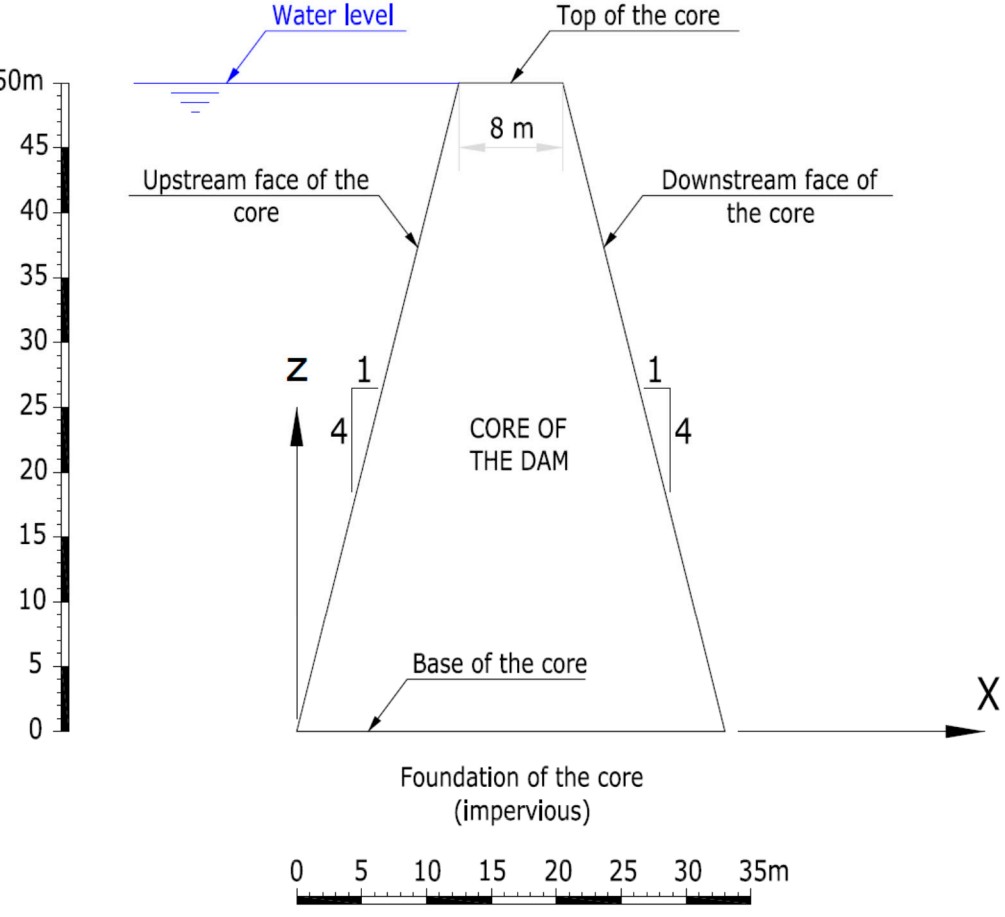

**Figure 1.** Two-dimensional problem with water flow within the dam cross section.

Therefore, in the plane of seepage, two different permeabilities can be defined in perpendicular directions. Generally, the vertical direction follows the compaction direction of the layers and has a considerably lower permeability than the horizontal direction. Despite this and although the flow that crosses the dam body from upstream to downstream may be affected, the seepage is generally controlled by the permeability of the soil in the horizontal direction since this is the predominant direction of the streamlines.

From these hypotheses and considering the boundary conditions, the Laplace formula that solves the problem of seepage through to a dam is formulated in position coordinates ($x, y$) by the following differential equation:

$$k_x \frac{\partial^2 \phi}{\partial x^2} + k_z \frac{\partial^2 \phi}{\partial z^2} = 0 \tag{1}$$

where $k_x$ and $k_z$ are the permeabilities in the $x$ and $z$ direction, respectively, and $\phi$ is the hydraulic potential at each point on the ground. From this equation and its boundary conditions, its hydraulic potential and, therefore, its pore pressure can be obtained at each point of the filtering ground. This is how the equipotential lines on the ground are defined, so that the flow takes place in the direction of the maximum potential gradient and, therefore, the current lines are orthogonally defined (Figure 2). With the routes of the flow lines, the area of soil that each equipotential line passes through can be determined and, together with the difference in value between equipotential lines ($\Delta\phi$), it is easy to obtain, using Darcy's law, the filtration speed and seepage flow.

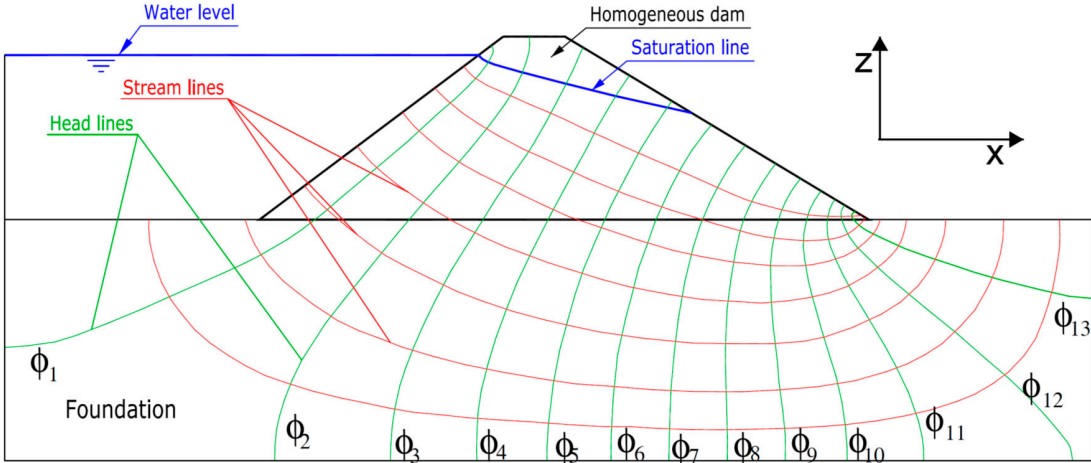

**Figure 2.** Equipotential lines and streamlines.

## 2.2. Finite Difference Method

### 2.2.1. Basis

Due to the assumed hypothesis of incompressibility during the filtration phenomenon, the flow modeling may be done on its own, independent of the usual mechanical calculation. Thus, Equation (1), along with those that define its boundary conditions, are solved using a finite-difference approach based on a discretization of the medium into zones of two overlays of triangles.

In the finite difference method, every derivative in the set of governing equations is replaced directly by an algebraic expression written in terms of the field variables at discrete points in space; these variables are undefined within elements. Using the approach by Wilkins [24], boundaries can be any shape and any element can have any property value.

An explicit, time marching method to solve the algebraic equations was used in the present study. Although a static solution to the problem has to be solved, the dynamic equations of motion were included in the formulation. One reason for doing this was to ensure that the numerical scheme was stable when the physical system being modelled was unstable. In contrast, schemes that do not include inertial terms must use some numerical procedure to treat physical instabilities.

### 2.2.2. Finite Difference Model

The study was carried out by developing computer numerical models, using a two-dimensional analysis of a dam core with trapezoidal geometry. These numerical models, applying different calculation hypotheses, were used to know the various water seepage paths that take place in the dam core as a consequence of the flow of water circulating through it.

The calculations used to determine the water seepage flow rates were carried out using the commercial software of finite differences FLAC 2D (fast Lagrangian analysis of continua) [25], which was developed by the Itasca company (Minneapolis, Minnesota, USA). This program is widely used in the study of geotechnical problems.

The FLAC numerical calculation software with the hypotheses set above can model fluid flow through permeable solids, such as soils. For the resolution of these flow problems, as indicated above, an uncoupled resolution was followed by only studying the phenomenon of water seepage regardless of the tensile-deformational state. Hence, it is not necessary to enter the resistant parameters of the materials or deformational characteristics.

The calculations were carried out using a simplified model of the dam (Figure 3), where only its core was modelled. This simplification was based on the starting hypothesis that the dam's upstream and downstream shells are absolutely permeable and only the core is the impervious element of the dam. Likewise, it was taken into consideration that the foundation of the core base is horizontal and behaves as an impervious element, which implies in practice that the foundation is much more impervious than the dam body. Therefore, the water flow only has the possibility of crossing the dam core from the upstream side to the downstream side.

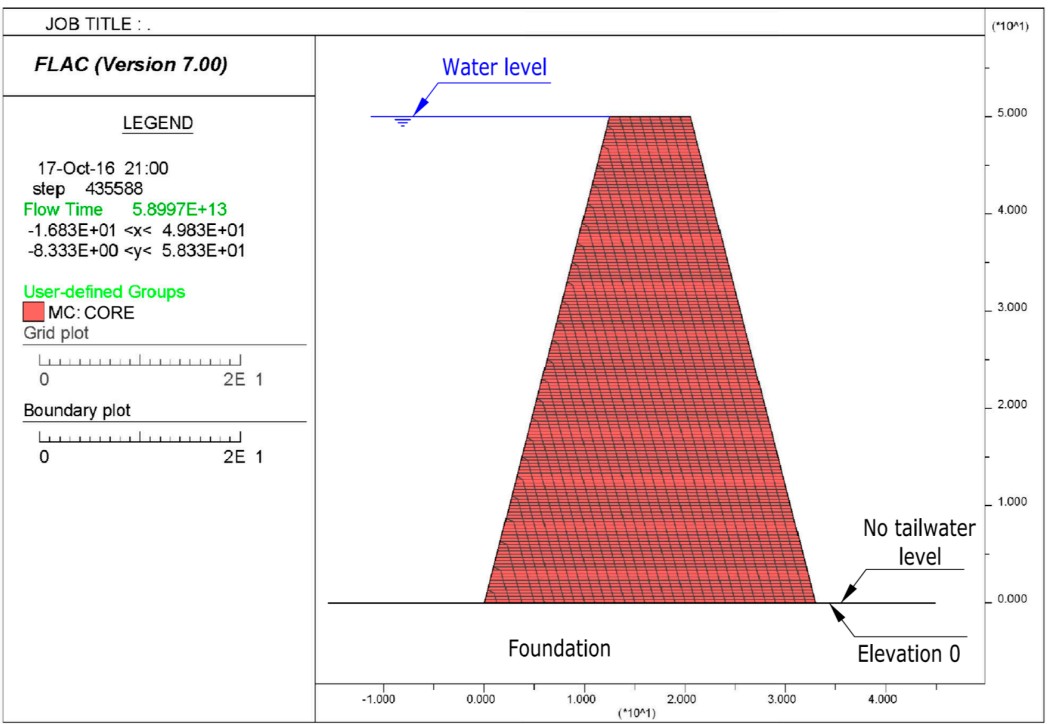

**Figure 3.** Numerical model of finite differences using FLAC 2D (fast Lagrangian analysis of continua).

In every calculation, the mechanical boundary conditions (Figure 3) were set for level 0 (base of the core) and the foundation ground of the core was non-deformable (restriction of movement in the horizontal and vertical directions in all the nodes located at level 0), while, for the rest of the nodes in the model, there were no imposed movement restrictions, which allowed them to move freely.

It was also necessary to define the hydraulic boundary conditions (Figure 3), with the upstream face of the core being permeable and the water level located. This is detailed in the following sections depending on the calculation hypothesis being considered, between the level $H$ (top of the core) and the level $0.6H$, where $H$ is the height of the dam core. As an additional hypothesis, it was also considered that the downstream face of the core was permeable and that, in addition, there was no tailwater level.

As mentioned in the introduction, two different types of analysis were carried out: (1) the effect of the zoning of the dam core by varying the height of the core, the inclination of its upstream and downstream faces, the value of the coefficient of permeability of the materials, and the height of water in the reservoir to study how they influence the seepage flow, as shown in Section 3; (2) the possibility of using as the core material a single heterogeneous soil with great granulometric variability and therefore permeability, as shown in Section 4.

2.2.3. Model Validation

The validation of the numerical model can be contrasted by checking the predictive capacity of the seepage flow in a dam where field measurements are available.

In particular, the Guadalcacín dam, located on the Majaceite river (Cádiz, Spain), was studied. The dam has a maximum height above foundations of 79 m, a crest length of 267 m, and a maximum reservoir capacity of 836 Hm$^3$. It is an earth dam with a straight plant and a central clay core with an estimated permeability coefficient of $10^{-5}$ cm/s. The crest of the core is at a height of 109 m and has a width of 7 m. The core has been built with 1H:4V slopes and is supported on rock made up of alternating limestone and marl, according to an impervious Jurassic outcrop in this area.

The standard section of the dam and a general photo are shown in Figure 4.

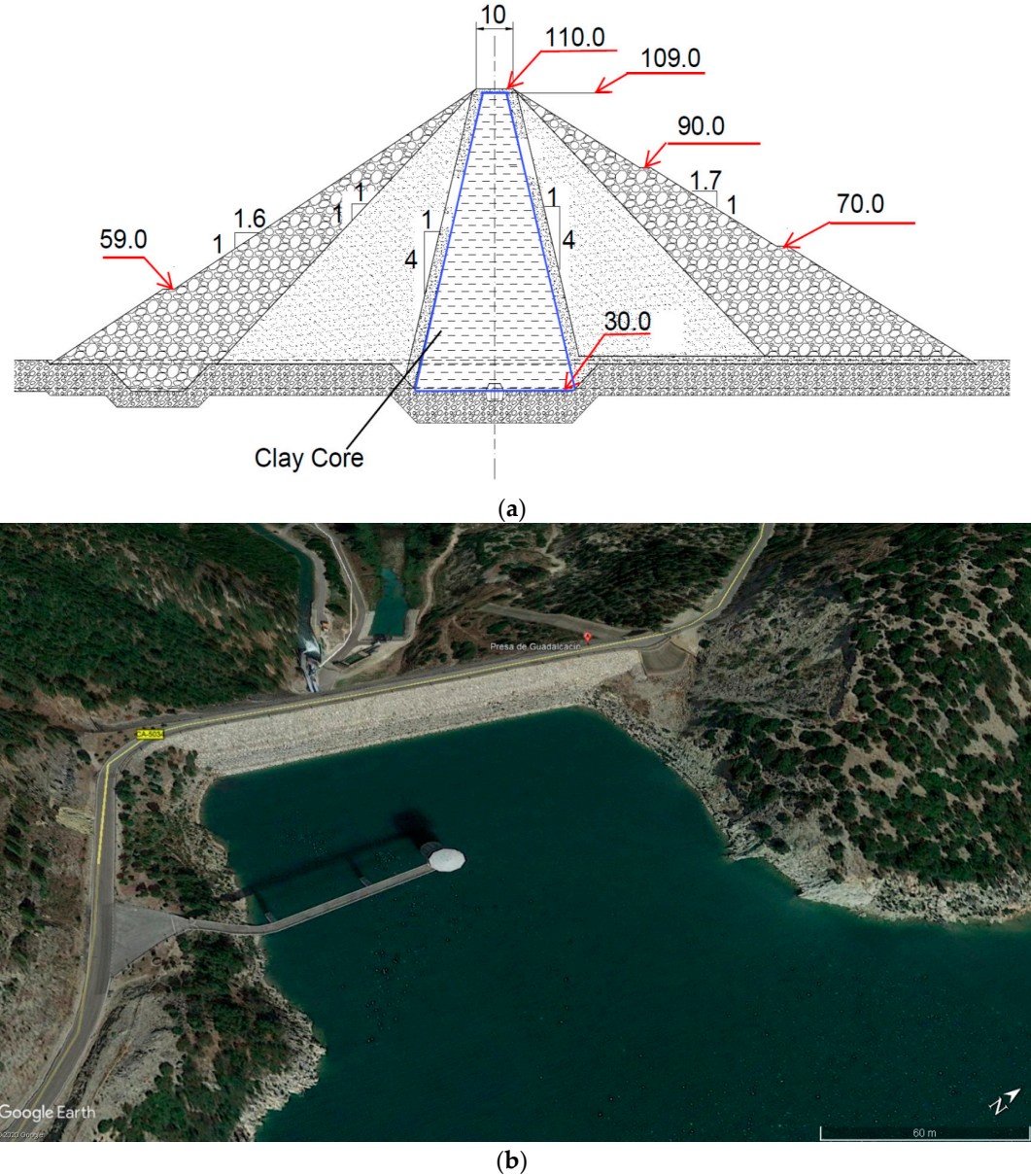

(**a**)

(**b**)

**Figure 4.** Guadalcacín dam: (**a**) standard section; (**b**) general photo (https://www.google.com/earth/index.html).

The Guadalcacín dam is equipped with an extensive monitoring system, which was installed during its construction. The available data on the seepage flow covers periods from 1997 to 2008 [26]

and are represented in Figure 5. As can be seen, the reality is always more complex than the theoretical models and there is a great dispersion of field measurements at the highest levels of the reservoir and between different years of measurement, mainly due to water seepage through the foundation. In fact, there have been subsequent actions to waterproof the foundation.

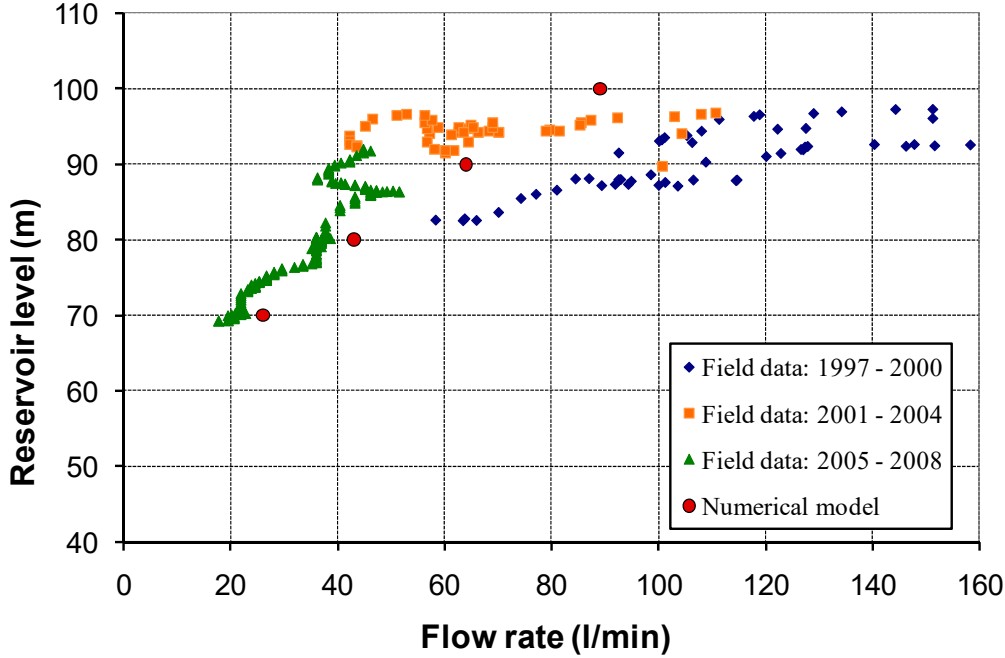

**Figure 5.** Flow rate results as a function of the reservoir level for the numerical model and field measurements at the Guadalcacín dam.

However, good results can be observed by comparing the field measurements with the results of the flow rate obtained in the numerical finite difference model using the FLAC 2D integrated over the entire length of the dam, which is also shown in Figure 5, and allows a reasonable validation of the calculation method used. For low reservoir levels (70 and 80 m) where no seepage through the foundation is expected, a very good fit was observed. However, for high water levels (90 and 100 m), the numerical results showed average flow values over the wide dispersion range of field flow measurements.

## 3. Core Zoning Study

### 3.1. Core Geometry

In every tested model, the core was assumed to be symmetrical and some variable parameters were set, regarding its geometry, core height, H, and the inclination of its faces, *i*.

Following the engineering practice for large dams, the height of the core was varied for $H = 20$ m, $H = 50$ m, and $H = 100$ m. Furthermore, the top width (C) of these modelled cores were established at $C = 6$ m, $C = 8$ m, and $C = 10$ m, respectively. Likewise, the inclination of the upstream and downstream faces of the core, expressed as the ratio of horizontal to vertical distance, varied at $i = 1{:}3$, $i = 1{:}4$, and $i = 1{:}5$.

### 3.2. Water Height

To account for different operational situations of the reservoir, various water heights, *h*, were modelled in the different calculations of the study. Five possible water heights were as follows: $h = H$, $h = 0.9\,H$, $h = 0.8\,H$, $h = 0.7\,H$, and $h = 0.6\,H$.

Figure 6 is a scheme of the core geometries and water heights considered in the calculations.

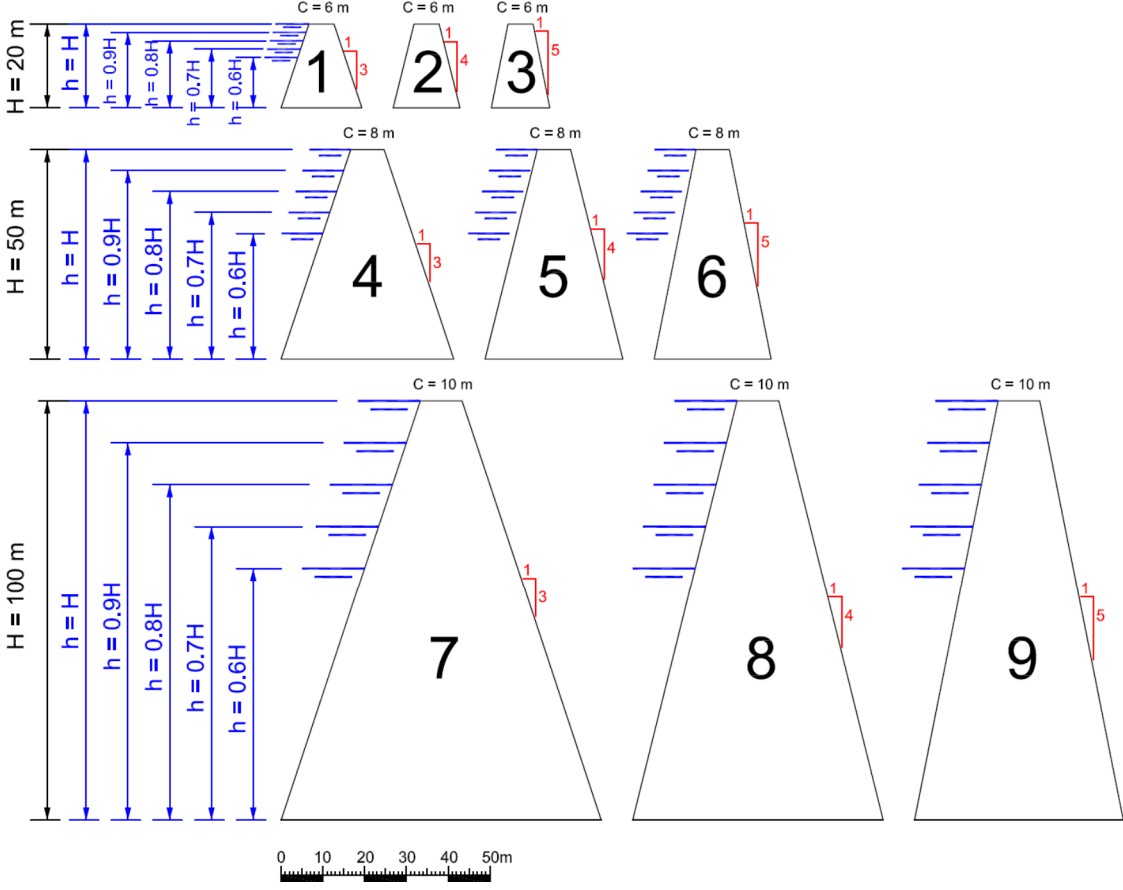

**Figure 6.** Cases studies analyzed for different cores and water heights.

### 3.3. Material Permeability

The material considered desirable as a dam core must have a maximum permeability of $10^{-5}$ cm/s, since above these values the material is considered semi-pervious (Hatanaka et al. [27] and Murray [28]). Therefore, with this mean reference value, other materials of lower and higher permeability are considered to be combined, according to different zonings, together with a filtration flow equal to or less than that corresponding to a single material of $10^{-5}$ cm/s. To have a wide range of variation of the permeability coefficients for this study, we considered five different values of the horizontal permeability ($k = k_x$) for the materials: $k = 10^{-3}$ cm/s, $k = 10^{-4}$ cm/s, $k = 10^{-5}$ cm/s, $k = 10^{-6}$ cm/s, and $k = 10^{-7}$ cm/s.

Likewise, two different hypotheses were considered on the permeability of the materials, i.e., calculations performed with an isotropic permeability distribution (horizontal permeability equal to the vertical permeability, $k_x = k_z = k$) and calculations carried out considering an anisotropic permeability distribution (horizontal permeability coefficient 10 times greater than the value of the vertical permeability coefficient $k_x = 10k_z$).

### 3.4. Materials Zoning

The core of the dam was zoned in different regions based on the different permeability values adopted for the materials in each calculation carried out. The various core zoning hypotheses that were adopted are shown in Figure 7.

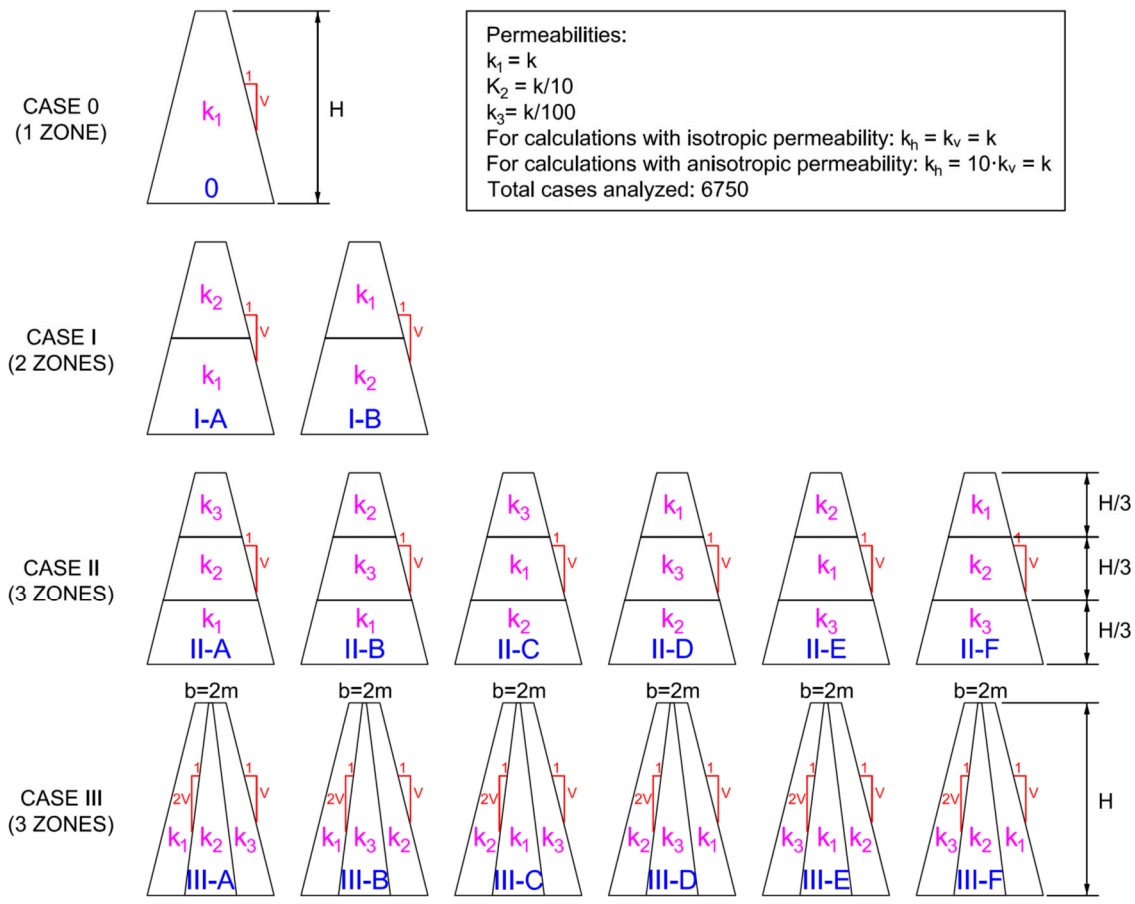

**Figure 7.** Different zoning core cases studies were analyzed according to the permeability of the material.

As can be seen, four different cases were taken for the core zoning: case 0, case I, case II, and case III, depending on how the materials with horizontal permeability $k_1$, $k_2$, and $k_3$ were set in the dam core, where for all cases it was $k_1 > k_2 > k_3$.

- Case 0 corresponded to a homogeneous core, without zoning, made up of a single material with horizontal permeability $k = k_1$.
- Case I showed a core dividing the height in half, i.e., 2 zones. Two sub-cases (I-A and I-B) were analyzed depending on whether the greater horizontal permeability and $k_1$, was considered in the upper half or in the lower half.
- Case II was similar to the previous one, but dividing the core height into three zones of identical sub-heights. Six subcases (II-A to II-F) were studied based on the distribution of horizontal permeabilities $k_1$, $k_2$ and $k_3$.
- Case III showed the zoning of the core in three parts, all of equal height. To carry out this zoning, a width of 1 m was taken in the central area of the top of the core and the central region was delimited with inclinations that were doubly inclined regarding the inclinations of the upstream and downstream faces of the core, as previously defined and indicated in Figure 7. In the same way as for case II, six subcases (III-A to III-F) were analyzed based on the distribution of horizontal permeabilities $k_1$, $k_2$ and $k_3$.

The different horizontal permeabilities that were considered in each zone were $k_1 = k$, $k_2 = k/10$ and $k_3 = k/100$ and thus: $k = k_1 \in [10^{-3}, 10^{-7}]$, $k_2 \in [10^{-4}, 10^{-8}]$, $k_3 \in [10^{-5}, 10^{-9}]$.

For each of the cases studied, the calculation program made it possible to obtain the water seepage paths in the core and, from it, the different values of the seepage flow were obtained.

### 3.5. Results

The combination of the different variations of the different parameters that were considered resulted in 6750 calculations. In particular, these parameters were the core height, inclination of the upstream/downstream faces of the core, height of water considered, values of the permeability coefficients for the materials, and zoning of the core.

The seepage flow through the core was obtained for all that enabled a sensitivity study to be carried out, analyzing how the variation of the parameters affected the values of the water seepage flow.

As stated above, the computer simulations were two dimensional calculations, so the resulting seepage flow was given per linear meter of the dam.

The obtained values of the seepage flows were within a very wide range, fundamentally depending on the different values of the coefficients of permeability that were adopted (Figure 8). The values of the seepage flow through the core of the dam were variable in the range 100 L/min (maximum value) and $10^{-5}$ L/min (minimum value) per meter of dam.

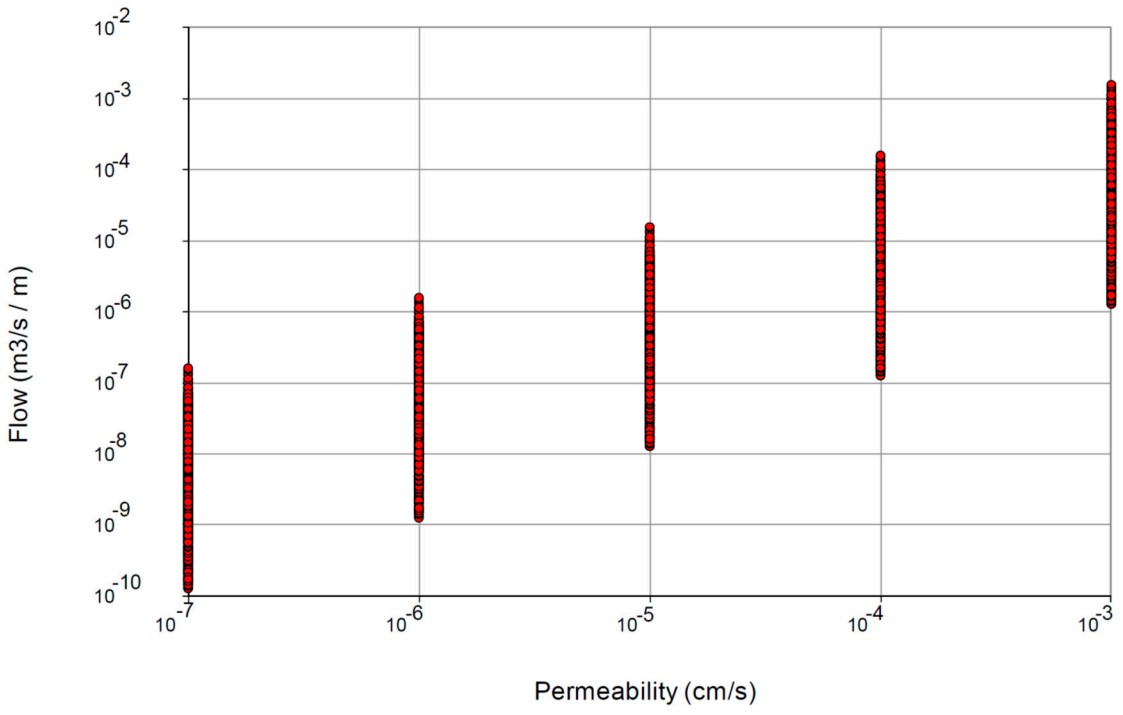

**Figure 8.** Values of the seepage flows through the core of the dam for all the calculation cases in relation to the value of the highest horizontal permeability.

In Figure 9, the different seepage flow values (normalized for the higher horizontal permeability $k$) obtained in the calculations are presented. Thus, the following considerations regarding zoning the core were deduced:

- The highest seepage flows were obtained when no zoning of the core was carried out (case 0). Then, the highest seepage flow were obtained for case I-A (lower half of the core with the highest permeability, $k_1$, and upper half of the core with the lowest permeability, $k_2 = k_1/10$) and case I-B (distribution of permeabilities at contrary to those adopted for case I-A), in that order.
- Case II obtained lower seepage flows than those obtained for cases 0 and I, resulting in the lowest seepage flows in cases II-D and II-F. Case studies in which the upper third of the core had the highest permeability coefficient, $k_1$, were assigned, and lower permeabilities were assigned to the lower two thirds, $k_2 = k_1/10$ and $k_3 = k_1/100$.

- Finally, the lowest seepage flow was obtained for calculation cases III and, more specifically, in cases III-A, III-B, III-C y III-E; the lowest permeability soil was not located in the downstream zone of the core.

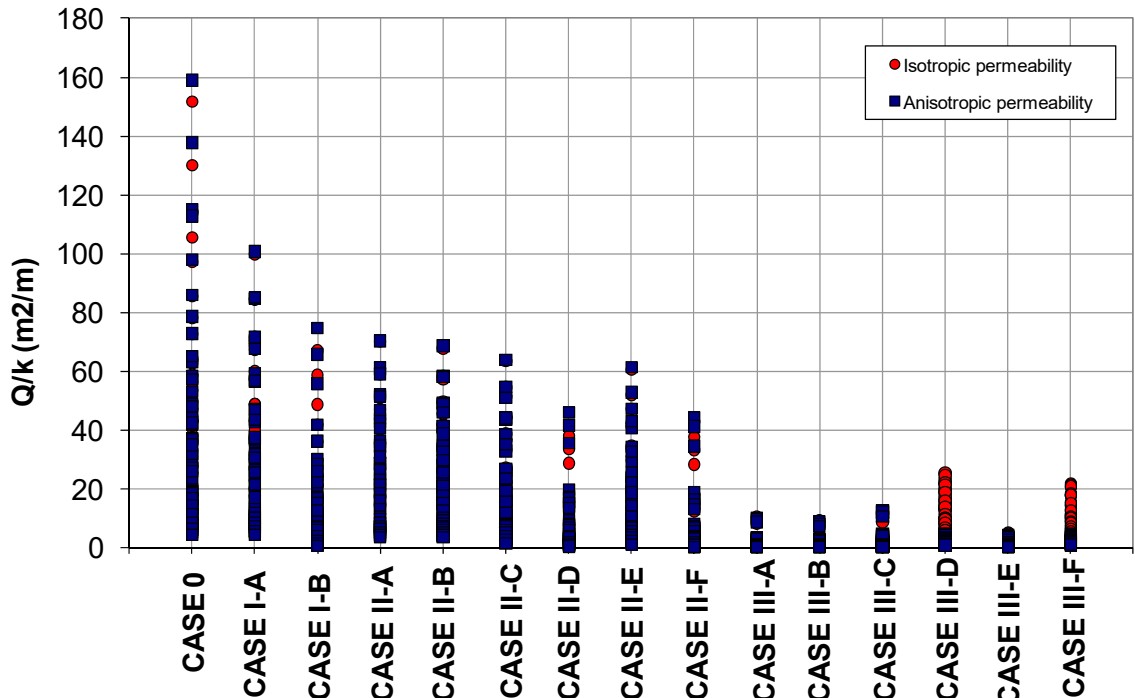

**Figure 9.** Seepage flow values normalized with respect to the highest horizontal permeability, identified by the analyzed cases studies.

Regarding the different dependency parameters considered in the models, it can be stated that:

- In general, similar flow rates were observed practically in all cases studied, regardless of whether an isotropic or anisotropic permeability distribution was considered. In Figure 10, the variation of the flow seepage is represented for the particular case of a 50 m height dam core verifying the little influence of the anisotropy in the variation range. Greater differences were only obtained in the calculation cases III-D and III-F, i.e., those in which the lowest permeability material was placed in the downstream face of the core (more detail given in the next section).
- Variations in the registered seepage flows were also observed depending on the geometry of the core. The seepage flow rates were higher for greater inclinations of the upstream and downstream core faces. Figure 10 shows the seepage flow variation as a function of the inclination of the upstream and downstream faces of the core for the particular case of $H = 50$ m dam core.
- The height of the water was also a factor taken into account for the resulting flow rates, logically obtaining higher seepage flow rates through the core for higher water heights. The influence for a maximum water height and a $0.60H$ water height is shown in Figure 10.



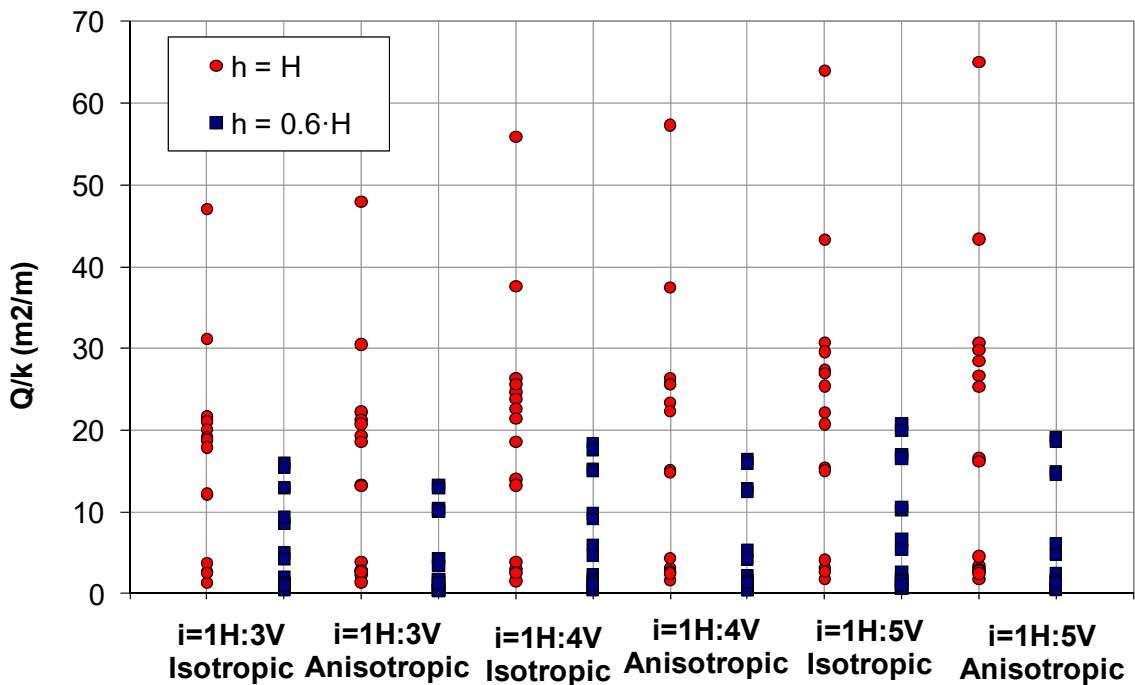

**Figure 10.** Values of the seepage flows normalized with respect to the highest horizontal permeability, as a function of the inclination of the upstream and downstream core faces for $H = 50$ m and maximum water heights and $0.6H$.

### 3.6. Discussion of Results

#### 3.6.1. Case 0: Only One Material

For a proper interpretation of the results and to determine the influence different parameters have on the seepage flow, a representation was made for each case where the abscissa axis is the height of the water in the reservoir and the ordinate axis is the normalized flow relative to what we call modified maximum flow ($Q_{max,m}$). Logically, the maximum seepage flow ($Q_{max}$) corresponds to the case of maximum reservoir water height and the modified maximum flow is defined as: $Q_{max,m} = Q_{max} \cdot (h/H)$. In Figure 11, this representation is shown for case 0, which incorporates every calculation, varying the height of the dam core, the height of water, the inclination of the upstream and downstream core faces and the permeabilities, and in both isotropic and anisotropic cases. In this case 0, the maximum seepage flow was $Q_{max} = 1.59$ L/s and corresponds to $h = H = 100$ m, $i = 1: 5$ and anisotropic permeability.

The following interpretation of the results can be made from Figure 11:

- The anisotropy of the permeability hardly exerts an influence because the flow is preferably horizontal, with the value of the horizontal permeability the one that controls the seepage. It can be seen that the adjustment curves are very close to each other and have a high correlation, regardless of the anisotropy.
- The higher the water level in the reservoir, the more seepage flow is filtered, obtaining in the normalized representation of Figure 11 a practically linear trend.
- All the results of the normalized representation are approximately aligned in a single curve regardless of the height of the dam core, the inclination of the upstream and downstream faces of the core, the anisotropy and the permeability value.

Therefore, the seepage flow corresponds to the case of maximum reservoir water height ($Q_{max}$) and thus it is possible, in general, to obtain the flow through an earth dam core with a homogeneous material for any dam height ($H$), water level ($h$), slope inclination, permeability coefficient, and anisotropy

between horizontal and vertical permeability. First, the modified maximum flow is calculated as: $Q_{max,m} = Q_{max} \cdot (h/H)$ and the seepage flow (Q) is obtained using Figure 11.

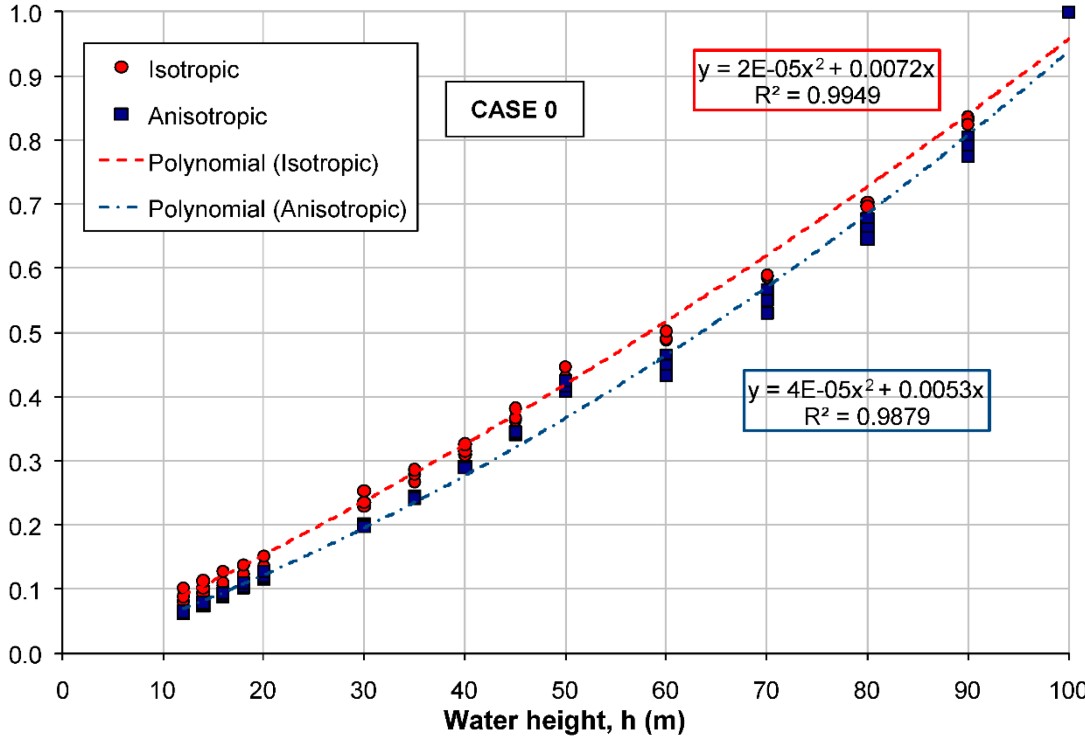

**Figure 11.** Normalized seepage flow with respect to the modified maximum seepage flow ($Q_{max,m}$) of the parametric study as a function of the water height in the reservoir for case 0.

### 3.6.2. Case I: Two Materials Set Horizontally

A similar representation can be made for case I (Figure 12). In this case, the more water height is in contact with the permeable material ($k_1$) in relation to the impermeable one ($k_2$), the more seepage flow. In particular, when the water height is equal to the dam core height, the two subcases I-A and I-B have the same height in contact with water for the two types of soil and, therefore, the same normalized seepage flow is obtained. ($Q/Q_{max,m}$) in both subcases (zones "1", "2", and "3" in Figure 12). For water heights under the height of the dam core, the normalized seepage flow ($Q/Q_{max,m}$) is greater in subcase I-A than in I-B because the decrease in the water height affects to the most impermeable soil. Moreover, as can be seen in Figure 12, for the full reservoir ("1", "2" and "3"), the same results were obtained in case I and in case 0 in relative terms (in relation to $Q_{max,m}$), but obviously in absolute terms in case 0 the seepage flow will be greater than in case I for any situation.

A representation adapted to two different types of soil can be made, with different permeability. In this case, we introduced the factor $\alpha_I = h_{k2}/h_{k1}$ that takes into account the proportion of the soil heights in contact with the water in the reservoir, between the material with the lowest permeability ($k_2$) and the one with the highest permeability ($k_1$). This factor must always be less than or equal to 1. The subscript "I" indicates that the factor corresponds to case study I. Therefore, to generalize, a factor for the homogeneous case $\alpha_0 = 1$ could be defined:

$$\alpha = \begin{cases} \textbf{Case 0}: \ \alpha_0 = 1 \\ \textbf{Case I}: \ \alpha_I = \frac{h_{k2}}{h_{k1}} = \frac{2h}{H} - 1, \ \alpha_I \leq 1 \end{cases}$$

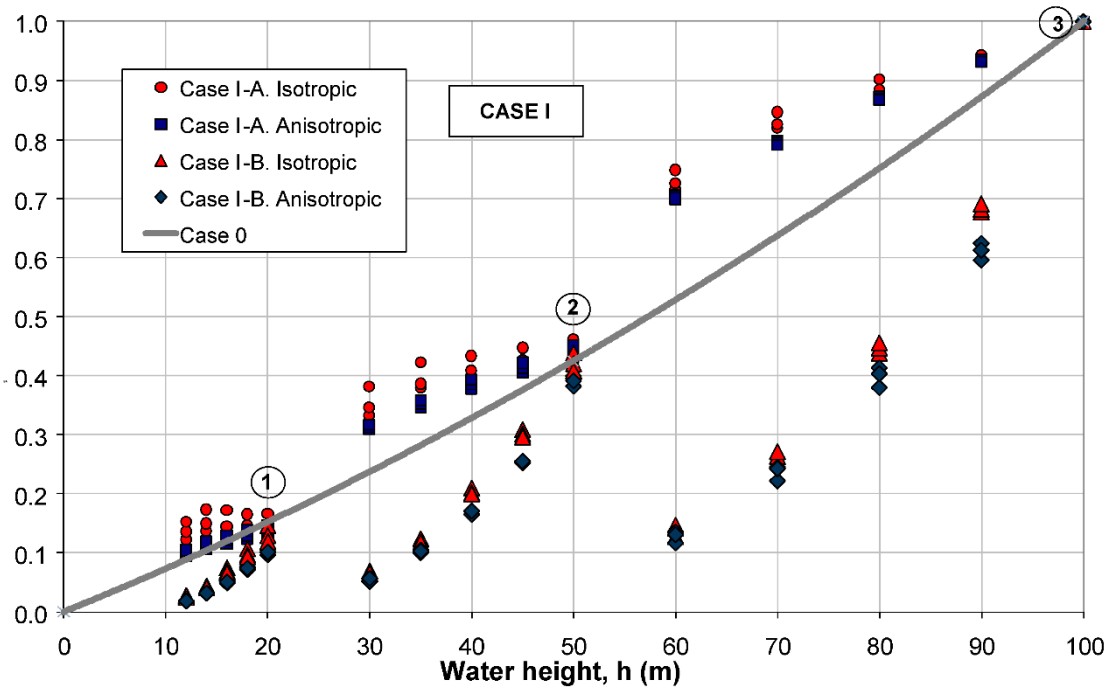

**Figure 12.** Normalized seepage flow with respect to the modified maximum seepage flow ($Q_{max,m}$) of the parametric study as a function of the water height for case I.

In this way, the so-called normalized general seepage flow ($Q^*$) affected by coefficient $\alpha$ is represented, so that $Q^* = Q \cdot \alpha / Q_{max,m}$, with respect to the height of water in Figure 13, where the following assessments can be done:

- The anisotropy of the permeability hardly has any influence because the flow is preferably horizontal.
- The higher the water height, the more seepage flow is produced, thus obtaining a practically linear trend in the normalized representation of Figure 13.
- The incorporation of the coefficient $\alpha$ allows the subcases I-A and I-B to be aligned in the same 3 curves, corresponding to the dam core heights of 20 m, 50 m, and 100 m.
- Following the evolution of the curves in case I, points "a" and "b" can be extrapolated to correspond to a water height $h = 20$ m for the dam core of $H = 50$ m, and to a water height of $h = 50$ m for the $H = 100$ m dam core, respectively. In these cases, it can be observed that the evolution of the curves moves away from the normalized general seepage flow predicted in the homogeneous case as the height of the water in the dam decreases. Specifically, points "a" and "b" correspond to all the material that is homogeneously more impermeable ($k_2$) and, therefore, regarding case 0 (of normalized general seepage flow $Q_0^*$), where the soil has homogeneous permeability $k_1$, a seepage flow ($Q_1^*$) in the ratio $k_2/k_1 = 0.1$, as can easily be observed in Figure 13.
- The zoning corresponding to subcase I-B is more efficient, as it reduces the seepage flow, although it requires about twice as much low permeability material as in subcase I-A.
- The values represented in Figures 12 and 13 correspond to normalized seepage flows. To make an analysis of absolute seepage flows, it is necessary to refer the values to maximum flows. In case I, the maximum seepage flow is $Q_{max} = 1$ L/s for subcase I-A and $Q_{max} = 0.75$ L/s for subcase I-B, and they correspond to $h = H = 100$ m, $i = 1:5$ and anisotropic permeability.

Therefore, the seepage flow corresponds to the case of maximum reservoir water height ($Q_{max}$) and it is possible, in general, to obtain the flow through an earth dam core with two materials "1" and "2" set horizontally for horizontal permeabilities $k_1$ and $k_2 = k_1/10$, for any dam height ($H$), water level ($h$), slope inclination, permeability coefficient, and anisotropy between horizontal and vertical

permeability. First, the modified maximum flow ($Q_{max,m}$) and the factor $\alpha$ are calculated as: $Q_{max,m} = Q_{max} \cdot (h/H)$ and $\alpha = 2h/H - 1$ (with $\alpha \leq 1$), and, finally, the seepage flow ($Q$) is obtained using Figure 13.

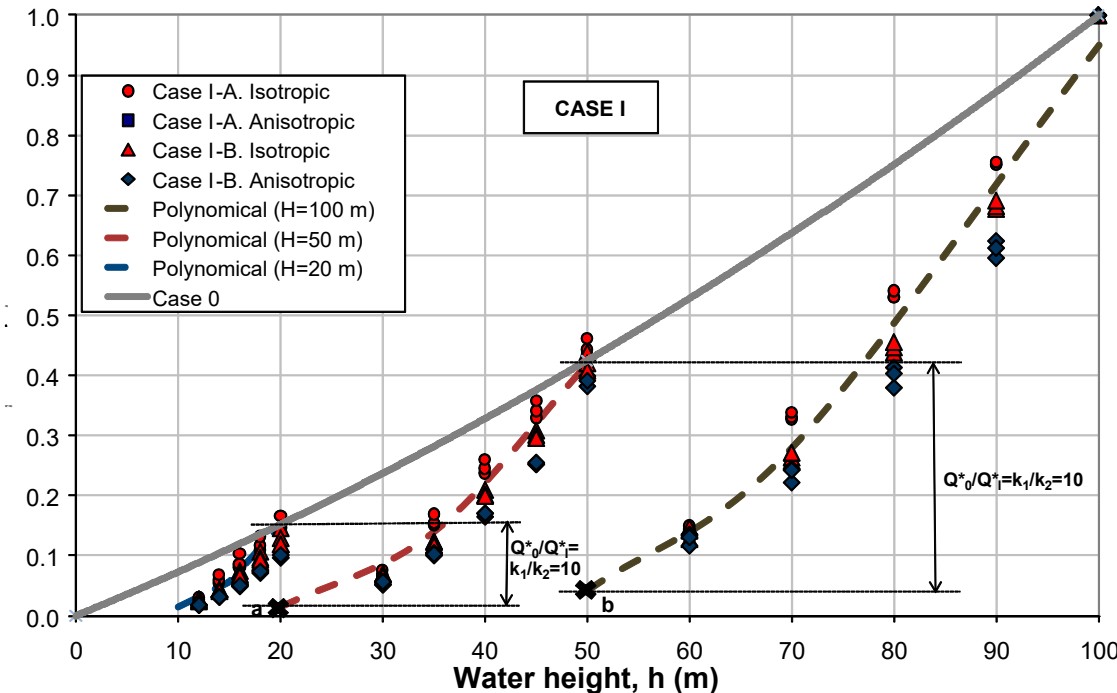

**Figure 13.** Normalized general seepage flow $Q^* = Q \cdot \alpha / Q_{max,m}$ of the parametric study as a function of the water height for case I.

### 3.6.3. Case II: Three Materials Set Horizontally

After carrying out the normalized representation regarding the previously defined modified maximum seepage flow ($Q/Q_{max,m}$) for case II and three horizontal soil layers of different permeabilities (Figure 14), we can make the following considerations:

- The anisotropy of the permeability hardly has any influence because the seepage flow tends to be horizontal.
- The higher water height is than the more seepage flow is produced, resulting in the normalized representation of Figure 14. Thus, there is a practically linear trend in subcases II-A, II-B, II-C, and II-E, as well as an exponential trend in the II-D and II-F subcases.
- The permeable layer of soil controls the seepage paths, so that if this more permeable soil is below, the order of the two materials above hardly have any influence. Thus, in subcases II-A and II-B, similar results were obtained.
- When the flow stream lines are controlled by the lower soil because it is the most permeable, as in case I, a factor $\alpha$ can be applied to transfer these subcases (II-A and II-B) to that corresponding to a lower impermeable soil, where the upper seepage paths are controlled by the intermediate soil when it is more permeable than the upper soil (corresponding to sub-cases II-C and II-E). In case II, the factor can be defined by $\alpha_{II}$ for the case of three materials, so that in general:

$$\alpha = \begin{cases} \textbf{\textit{Case 0}}: \ \alpha_0 = 1 \\ \textbf{\textit{Case I}}: \ \alpha_I = \frac{2h}{H} - 1, \ \alpha_I \leq 1 \\ \textbf{\textit{Case II}}: \ \alpha_{II} = \frac{1}{2}\left(\frac{3h}{H} - 1\right), \ \alpha_{II} \leq 1 \end{cases}$$

In this way, as in case I, the normalized general flow ($Q^*$) affected by the coefficient $\alpha$, regarding the water height, is represented in Figure 15.

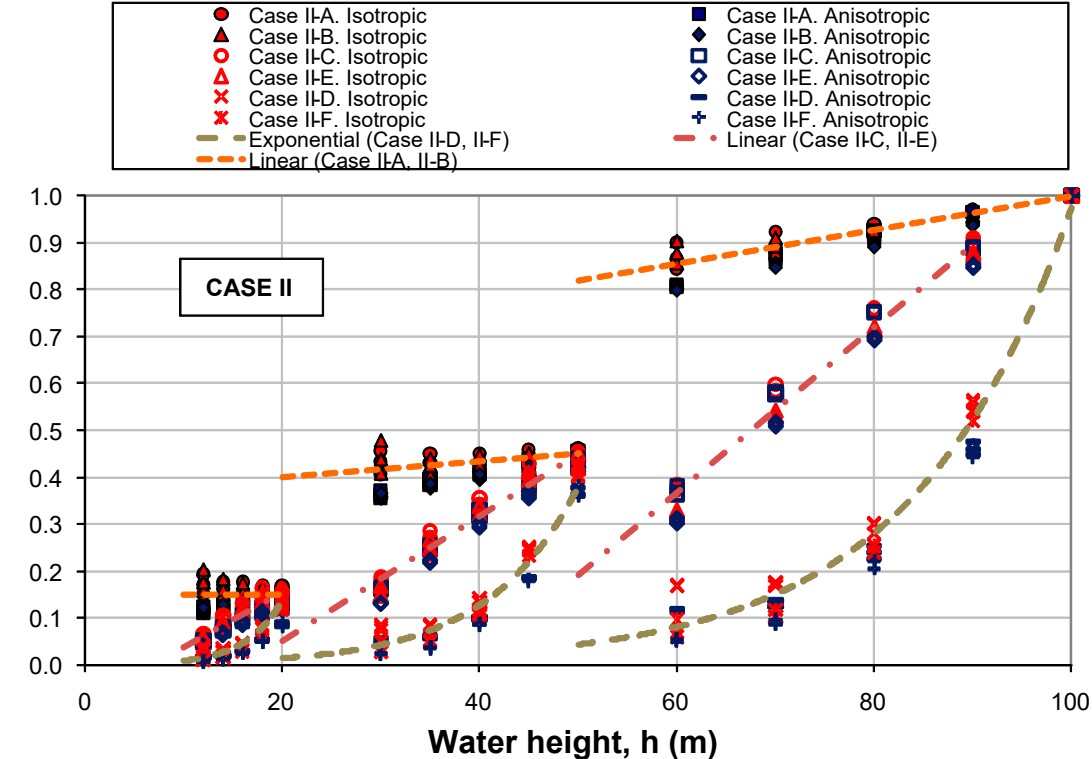

**Figure 14.** Normalized seepage flow with respect to the maximum modified seepage flow ($Q_{max,m}$) of the parametric study as a function of the water height for case II.

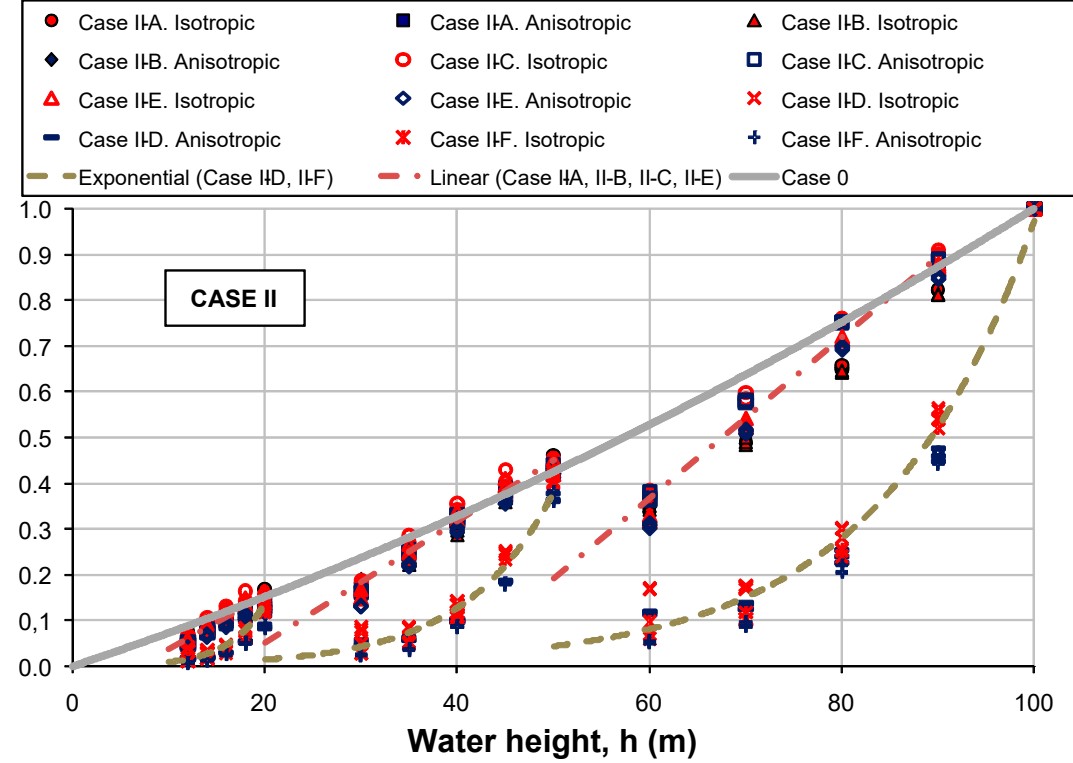

**Figure 15.** Normalized general seepage flow $Q^* = Q \cdot \alpha / Q_{max,m}$ of the parametric study as a function of the water height for case II.

- The seepage flow corresponding to the parametric study of case 0 is represented in Figure 15, which clearly shows that it corresponds to an upper envelope of all the results of case II.
- As shown in Figures 14 and 15, the most effective zoning of the core corresponds to subcases II-D and II-F, where the most permeable soil is set at the top. This configuration greatly reduces seepage flow rates.
- Once again, attention is drawn to the fact that the values represented in Figures 14 and 15 correspond to normalized seepage flows. To make an analysis of absolute seepage flows it is necessary to refer to the values of maximum seepage flows. In case II, the maximum seepage flow is $Q_{max} = 0.7$ L/s for subcase II-A, $Q_{max} = 0.69$ L/s for subcase II-B, $Q_{max} = 0.64$ L/s for subcase II-C, $Q_{max} = 0.46$ L/s for subcase II-D, $Q_{max} = 0.62$ L/s for subcase II-E, and $Q_{max} = 0.44$ L/s for subcase II-F. All of them correspond to $h = H = 100$ m, $i = 1: 5$, and anisotropic permeability.

Therefore, the seepage flow corresponds to the case of maximum reservoir water height ($Q_{max}$) and thus it is possible to obtain the flow through an earth dam core with three materials: "1", "2", and "3", which are set as horizontal permeabilities $k_1$, $k_2 = k_1/10$, and $k_3 = k_1/100$ for any dam height ($H$), water level ($h$), slope inclination, permeability coefficient, and anisotropy between horizontal and vertical permeability. First, the modified maximum flow ($Q_{max,m}$) and factor $\alpha$ are calculated as: $Q_{max,m} = Q_{max} \cdot (h/H)$ and $\alpha = 1.5h/H - 0.5$ (with $\alpha \leq 1$), and, finally, the seepage flow ($Q$) is obtained using Figure 15.

### 3.6.4. Case III: Three Materials Set Vertically

To interpret the results, the representation of the normalized seepage flow regarding the maximum modified flow ($Q/Q_{max,m}$) was also made for case III, as presented in Figure 16. In this figure, three different zones are distinguished corresponding to the different analyzed sub-cases, where the most effective zoning that minimizes seepage flow corresponds, for water height of 90% or less, to sub-cases III-A and III-C, in which the most impervious material was in the last position. However, in these subcases, the seepage flow increased exponentially for a maximum water height situation and its seepage flow rates were comparable, and even higher, than those obtained in subcases III-B and III-E. This has been shaded and marked with a dashed line in Figure 16. In addition, the following considerations can be made:

- The effect of anisotropy is clearly appreciated in cases where the intermediate permeability material ($k_2$) is in the last position, since flows with a greater vertical component are generated in the transitions between materials.
- The efficiency of the core zoning is directly related to the permeability of the last vertical layer since, in case III, the entire seepage flow passes through all the materials, but the greatest loss of water head always corresponds to the last layer that cancels the hydraulic potential. Therefore, the more effective the flow in this last stratum than the lower the seepage flow rate. The most favorable situation corresponds, as mentioned, to subcases III-A and III-C, with the most impervious material at the end. However, this situation changes abruptly for maximum water height, where the seepage flows exceed those corresponding to intermediate subcases III-B and III-E with the intermediate permeability soil ($k_2$) positioned at the end. In the event that the soil with the highest permeability ($k_1$) is located in the last position, it corresponds to the situation of maximum seepage flow (subcases III-D and III-F), as is shown in Figure 16.
- The values represented in Figure 16 correspond to normalized seepage flows. To make an analysis of absolute seepage flows it is necessary to refer to the values of maximum seepage flows. In case III, the maximum flow is $Q_{max} = 0.1$ L/s for subcase III-A, $Q_{max} = 0.09$ L/s for subcase III-B, and $Q_{max} = 0.05$ L/s for subcase III-E. All of them correspond to $h = H = 100$ m, $i = 1:5$, and isotropic permeability; $Q_{max} = 0.13$ L/s for sub-case III-C of $h = H = 100$ m, $i = 1:5$, and anisotropic permeability. $Q_{max} = 0.25$ L/s for subcase III-D and $Q_{max} = 0.22$ L/s for subcase III-F corresponds to $h = H = 100$ m and $i = 1:4$ and isotropic permeability, respectively.

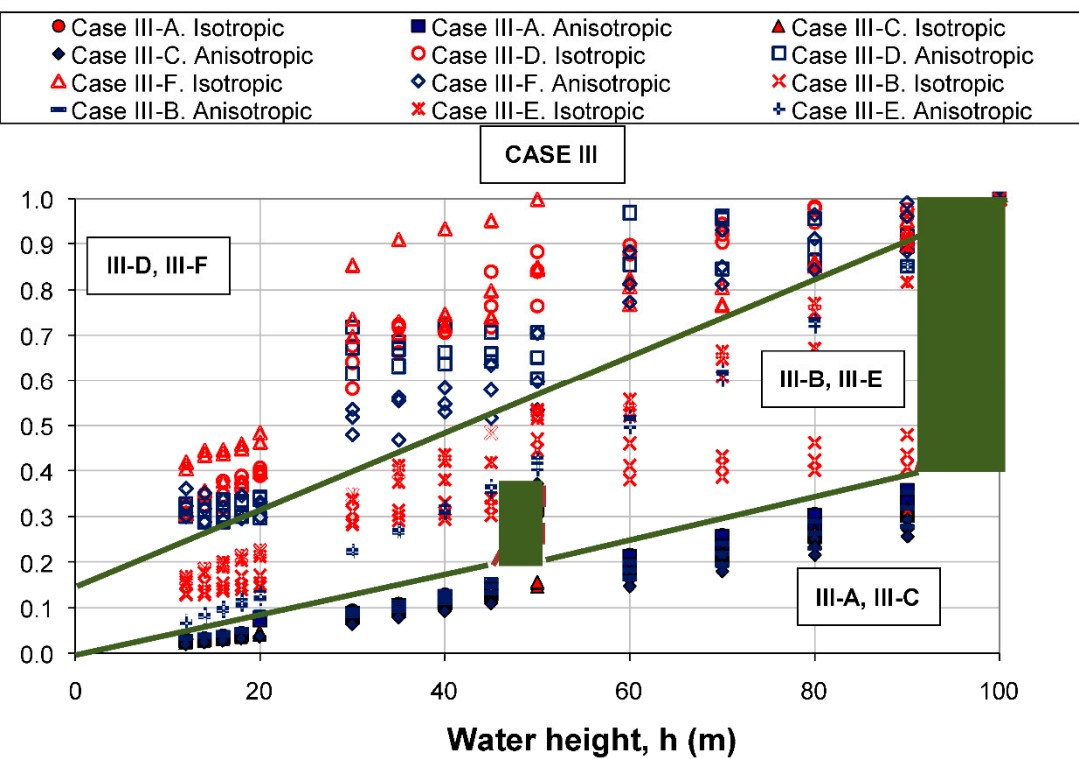

**Figure 16.** Normalized seepage flow with respect to the modified maximum seepage flow ($Q_{max,m}$) of the parametric study as a function of the water height for case III.

As indicated above, for maximum water heights of 90% or less, the seepage flow rate is drastically reduced in subcases III-A and III-C, with the smallest seepage flow, 0.033 L/s, corresponding to subcase III-A for $h = 90\%$.

When carrying out a comparative analysis of the zoning of cases 0, I, II, and III, it can be appreciated that:

- The maximum seepage flow rates are progressively smaller from case 0, case I, case II, and case III. In case III, minimum values were obtained with $Q_{max} \in (0.05 \text{ L/s}, 0.25 \text{ L/s})$; in case II the values were $Q_{max} \in (0.44 \text{ L/s}, 0.70 \text{ L/s})$; in case I the values were $Q_{max} \in (0.75 \text{ L/s}, 1 \text{ L/s})$; and finally, the highest values of maximum seepage flow corresponded to case 0 with values that reach 159 L/s.
- Attention should be drawn to the fact that the volume of all materials in case III were approximately the same, thus minimizing the environmental and economic impact. Whereas in case II and case I, a higher proportion of the more impervious material was always needed to optimize the seepage flow.
- For case III, with water heights under 90% of the dam core height, the most effective zoning was obtained for subcases III-A and III-C. However, for maximum water height, the seepage flow increased exponentially in these two subcases, and the zoning that presents cases III-B and III-E was more effective for the maximum water height situation.

Therefore, the seepage flow corresponds to the case of maximum reservoir water height ($Q_{max}$), where it is possible to obtain the flow through an earth dam core with three materials "1", "2", and "3", which are set vertically for horizontal permeabilities $k_1$, $k_2 = k_1/10$, and $k_3 = k_1/100$ for any dam height ($H$), water level ($h$), slope inclination, permeability coefficient and anisotropy between horizontal, and vertical permeability. First, the modified maximum flow is calculated as: $Q_{max,m} = Q_{max}\cdot(h/H)$ and, finally, the seepage flow ($Q$) is obtained using Figure 16.

## 4. Study of Heterogeneous Material for the Dam Core

### 4.1. Dam Core Geometry

The dam core was assumed to be symmetrical, with inclinations of both upstream and downstream faces 1H:4V (usual value of the inclination for dam cores built with heterogeneous materials), and a core height of 50 m was considered. The top width of the core was established at 8 m and, therefore, the width of the base was 33 m.

Figure 17 presented below shows the geometry that served as the basis for the elaboration of the calculation model that was prepared.

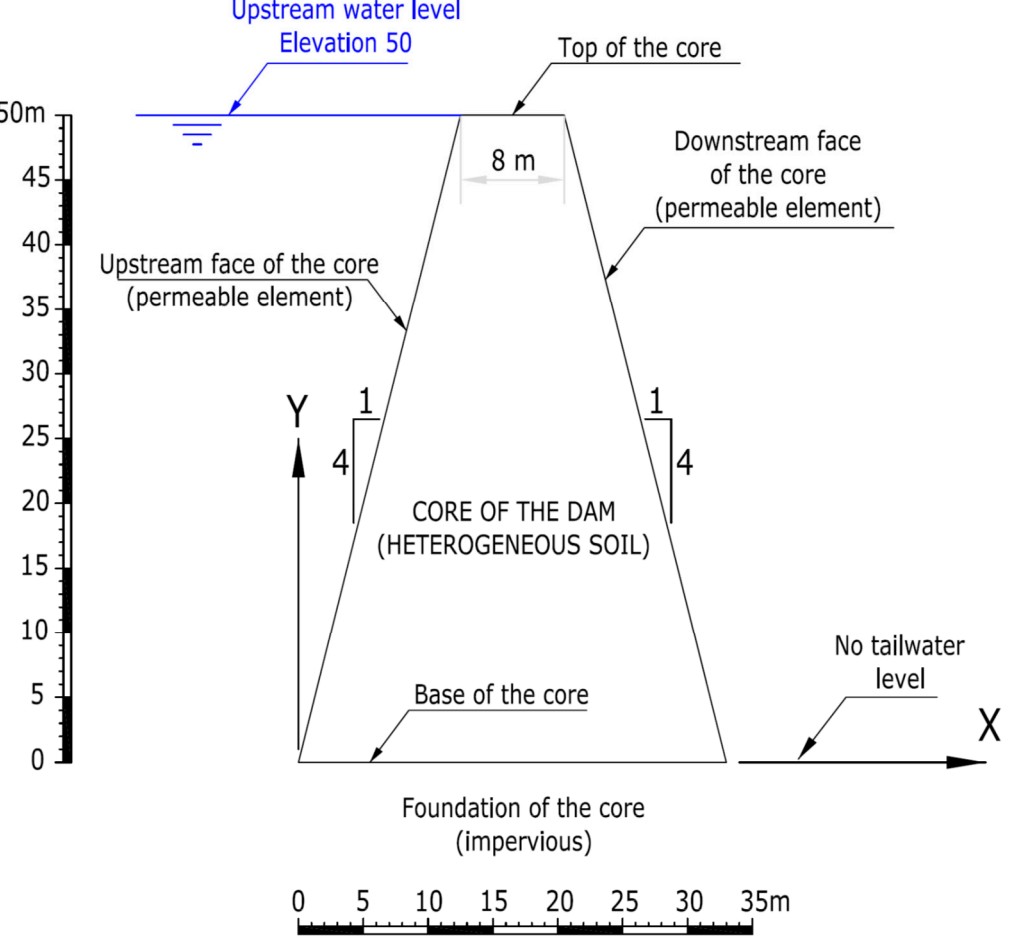

**Figure 17.** Calculation geometry of the core adopted for numerical modelling with heterogeneous material.

The origin of coordinates (X = 0, Y = 0) was arbitrarily taken at the heel of the core for all the calculations that were carried out.

### 4.2. Water Height

Regarding the hydraulic boundary conditions, it was considered that the upstream face of the core was permeable and that the water level was located at elevation 50 (top of the core). It was also considered that the downstream face of the core was permeable and, in addition, there was no tailwater level. The base of the core was considered as a waterproof level.

In Figure 17, in which the geometry of the model was shown, the adopted hydraulic boundary conditions were also included.

### 4.3. Permeability of the Material

In general, a heterogeneous material shows highly variable properties and its characterization presents significant uncertainties. These materials have a very extensive granulometry, with the presence of particles of very different sizes and permeabilities that vary over a wide range. In this research, heterogeneous material for seepage analysis is defined as soil that presents a great dispersion of permeability. Specifically, in this section, for the calculations carried out in this investigation, an average permeability value of $10^{-6}$ cm / s and a variation of the permeability of the material between $6.15 \cdot 10^{-14}$ cm/s of minimum value and $5 \cdot 10^{-3}$ cm/s maximum value is considered, as explained later.

The simulation of the construction of the core was carried out by means of a lift measured 4 m wide and 0.33 m high. In this way, the prepared numerical model had a total of 768 lifts. Each lift was made up of four elements of numerical discretization.

The assigned permeabilities were considered anisotropic. For each lift, it took a horizontal permeability coefficient value that was 10 times greater than the value of the vertical permeability coefficient ($k_h = 10 k_v$).

Since the dam core was built with a heterogeneous material in order to lift the model, we assigned a different permeability provided that the set of 768 adopted permeability coefficients adjusted to a lognormal distribution function. To define this permeability distribution function, its two characteristic parameters must be known: its mean value and its standard deviation. To do this, we proceeded as follows:

The value of the horizontal mean permeability was set to $k_h = 10^{-6}$ cm/s, a value considered as characteristic for a soil with which to build the dam core.

The standard deviation of the distribution function was obtained, indirectly, by fixing the dispersion of the permeability variable. For the same mean permeability value, the lognormal distribution function may be more "open" (greater permeability dispersion) or more "closed" (less permeability dispersion) depending on the standard deviation of the adopted value. The dispersion of the permeability will be given, therefore, by the values $Y_f$ and $Y_i$, the adopted extremes of the distribution function (Figure 18).

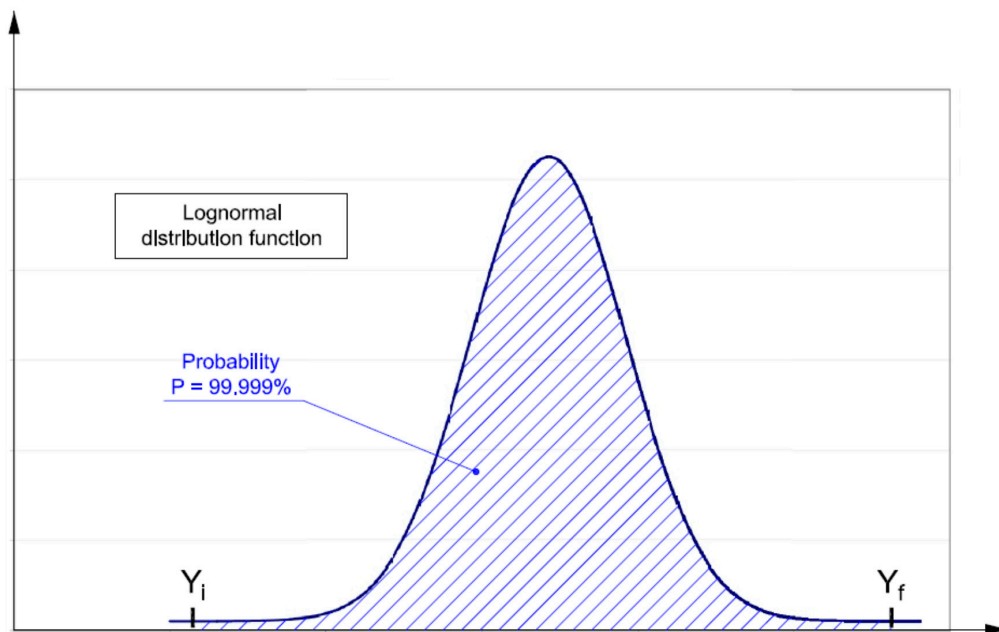

**Figure 18.** Maximum ($Y_f$) and minimum ($Y_i$) ends of the distribution function.

To fix the extreme values $Y_f$ and $Y_i$ of the lognormal distribution function, the concept of failure probability was introduced. Brown [29] indicated that the probability of failure for dams was around $p = 10^{-5}$. Thus, as a starting point, one of the extremes of the distribution function (the maximum,

$Y_f = 5 \cdot 10^{-3}$ cm/s) was fixed, then proceeded to calculate the minimum extreme, $Y_i$, applying the condition that the lognormal distribution function between $Y_f$ and $Y_i$ contains a probability of $(1-p)$, with $p = 10^{-5}$ as indicated above. In other words, it was stipulated that the defined distribution function must contain a probability between $Y_f$ and $Y_i$ of 99.999%. In accordance with these working hypotheses, $Y_i$ was fixed at $6.15 \cdot 10^{-14}$ cm/s.

### 4.4. Seepage Path and Streamlines

The seepage path was obtained for each distribution of permeabilities in the lifts of the dam core. From this, the seepage flowed through the core and the value of the maximum gradient occurred as a consequence of the circulating water flow. In normal practice, it is usually recommended that the seepage flow rates be limited below reference threshold values, usually set by the dam technicians.

The study of the gradient has the objective to show that the seepages running through the dam core do not produce an internal erosion phenomena. References like the dam project recommendations in Russia [30] indicate that the critical gradient may be around 5, which is why it is usual to recommend the construction of waterproof elements in dams with thicknesses greater than 20% of the water height.

For this reason, in the calculations, the maximum gradient was determined by identifying the area of the core (lift, height within the core of the dam, etc.) where it should be placed, following the hypotheses of work. It was assumed that the gradient could determine the design of the core and could be the one that occurs in some of the lifts that form the downstream face of the core. Moreover, it could be evaluated according to what is indicated in the diagram and in the expression shown in Figure 19, where $hp_i$ is the hydraulic potential in element "$i$", $d_{1\text{-}i}$ is the distance from the center of element 1 to the center of element "$i$", and $grad(T_i)$ represents the value of the gradient in the lift "$i$".

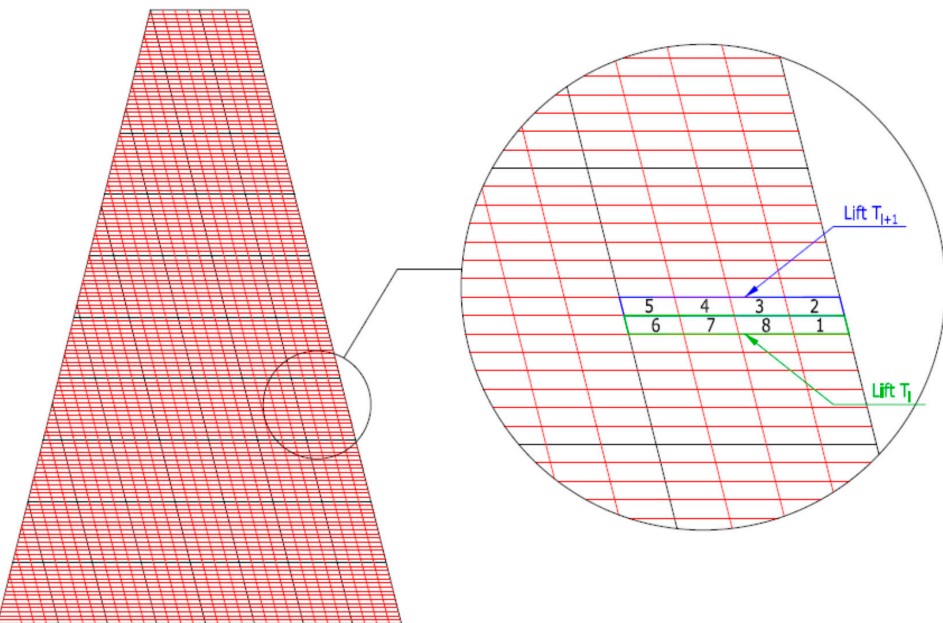

$$grad(T_i) = max \left[ \frac{hp_1 - hp_2}{d_{1-2}}, \frac{hp_1 - hp_3}{d_{1-3}}, \frac{hp_1 - hp_4}{d_{1-4}}, \frac{hp_1 - hp_5}{d_{1-5}}, \frac{hp_1 - hp_6}{d_{1-6}}, \frac{hp_1 - hp_7}{d_{1-7}}, \frac{hp_1 - hp_8}{d_{1-8}} \right]$$

**Figure 19.** Determination of the maximum gradient in calculations with a numerical model for a heterogeneous core.

Once the values of the 150 gradients (one for each of the 150 lifts located in the final 4 m of the downstream face of the dam core) were calculated, the maximum value of this gradient can be obtained, thus determining at what height of the dam core (in what lift) is produced.

### 4.5. Results and Discussion

A calculation routine introduced in the FLAC program was generated that can assign different permeability coefficients to the 768 lifts that constitute the core model. A condition was that the set of all these permeability coefficients adjust to a lognormal distribution function with mean horizontal permeability value of $k_h = 10^{-6}$ cm/s and dispersion of the horizontal permeability coefficients in the range $Y_f = 5 \cdot 10^{-3}$ cm/s and $Y_i = 6.15 \cdot 10^{-14}$ cm/s.

A Montecarlo analysis was thus generated with a total of 10,000 calculations carried out, obtaining in each of them the seepage path in the dam core, the seepage flow through the core and the maximum gradient produced in the core as a consequence of the circulating water flow.

The results of the 10,000 calculations are represented in various graphs for analysis. These graphs include the distribution of seepage flows through the core, the distribution of maximum gradients, the distribution of the height of the core where the gradients are at a maximum, the relationship between the seepage flow rates and the maximum gradients, etc. These graphs are shown below in Figure 20.

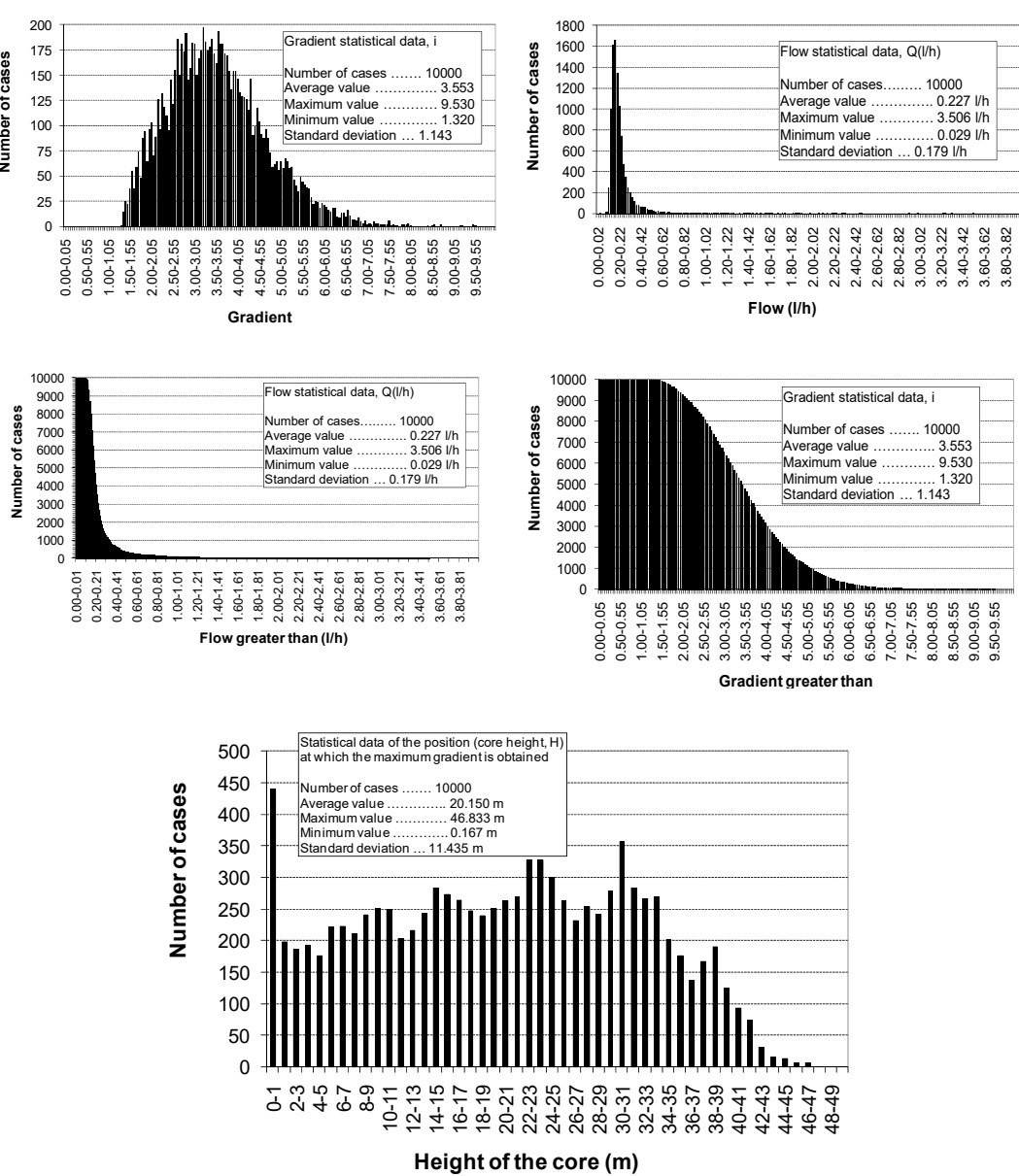

**Figure 20.** Results obtained with a numerical model for heterogeneous material.

Seepage flows are given per linear meter of dam (two dimensional calculations). The average value of the seepage flow obtained was around 0.23 L/h, with a maximum value of 3.51 L/h and a minimum value of 0.03 L/h, respectively. For approximately 90% of the calculations, the seepage flow through the core was under 0.30 L/h.

The maximum gradient was variable in the range 1.32 to 9.53, with an average value of 3.55. For approximately 90% of the calculations, the value of the maximum gradient in the core was under 5. Regarding the position at which the maximum gradients were detected, it can be indicated that they were observed at low heights, within the first few meters of the dam core height. These values were detected for moderate or low seepage flow rates, below 0.5–0.6 L/h.

Contrary to what one might initially think, it was observed that the values of the highest maximum gradients were not necessarily associated with the values of the highest seepage flow rates. High gradients were obtained for situations in which the seepage flow rates were moderate and low.

As the calculations were carried out and the maximum seepage flow rates and gradients were obtained, their mean values were found for the first 20, 50, 100, 200, . . . , 1000, 2000, . . . etc. calculations. This information is presented in Figure 21.

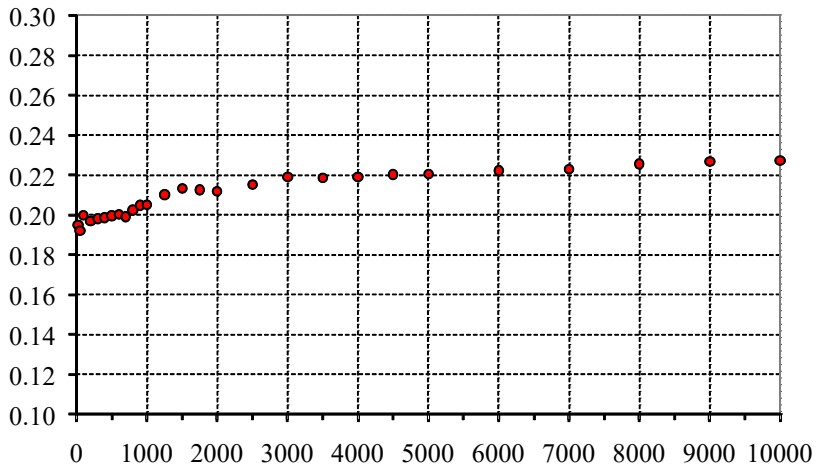

**Number of cases**

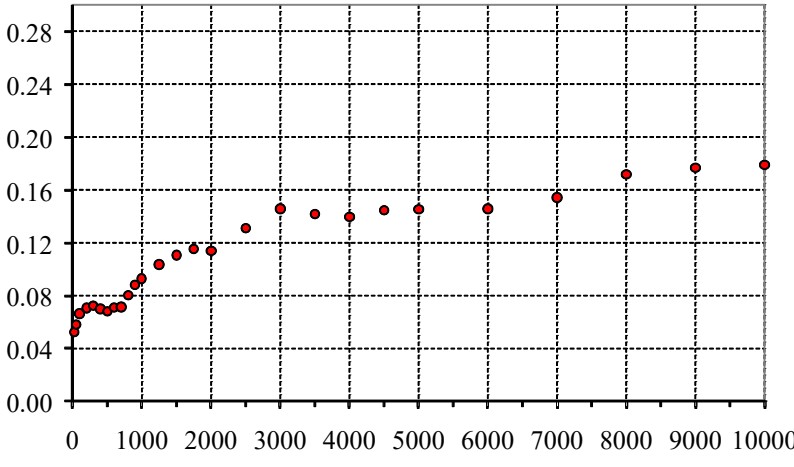

**Number of cases**

**Figure 21.** *Cont.*

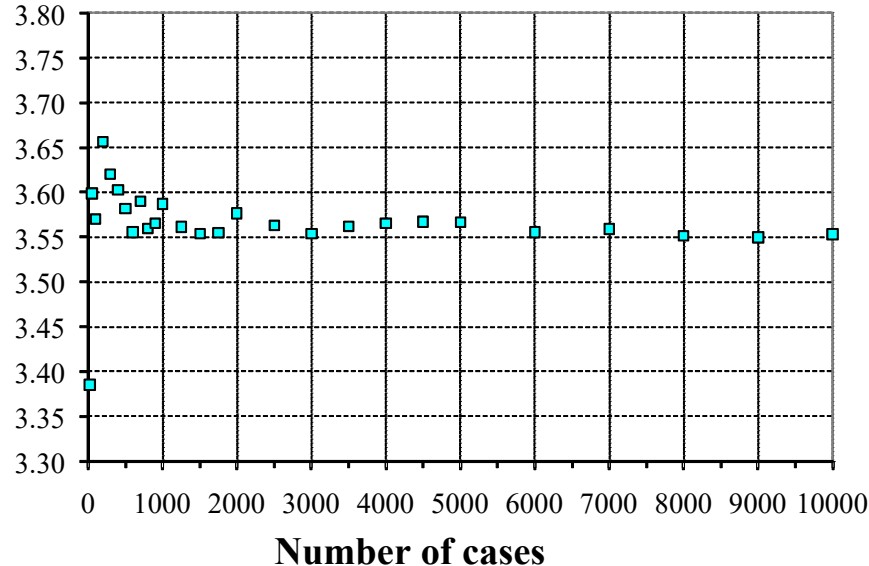

**Number of cases**

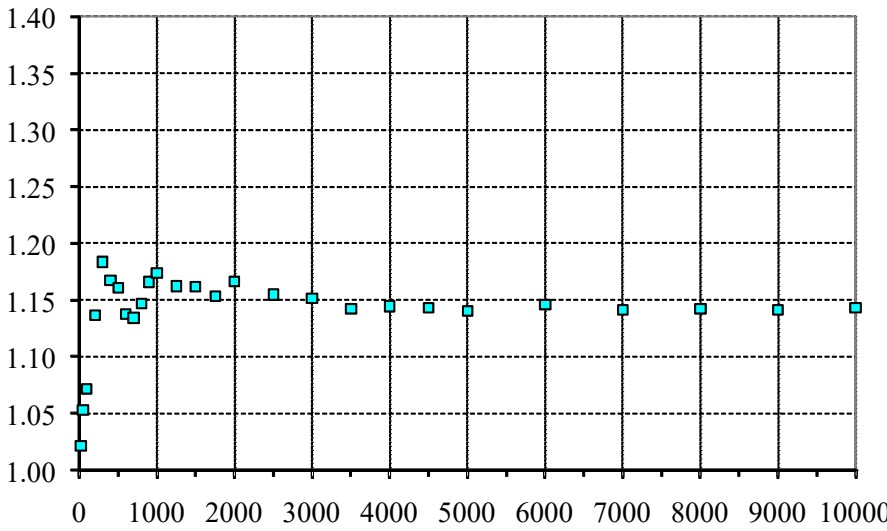

**Number of cases**

**Figure 21.** Partial average seepage flow rates and mean gradients in Montecarlo analysis for heterogeneous material.

It can be seen how, from a number of calculations greater than or equal to 3000, the average seepage flow was quite stable (0.22–0.23 L/h). However, we observed a slight tendency to increase. Regarding the value of the mean maximum gradient, it was quite stable for a number of calculations above 1000. The value of this mean maximum gradient was maintained with slight variations between 3.55–3.56. Therefore, from the Montecarlo study, the appropriate possibility of using heterogeneous material as a dam core can be concluded since:

- The seepage flow rates are limited to sufficiently low values when the average value of the permeability is $10^{-6}$, despite the great dispersion of the material. In this regard, it can be compared with the seepage flow rates obtained for the analysis with homogeneous materials indicated in Section 3. In this case, permeabilities between $10^{-3}$ and $10^{-7}$ were used. If we compare the result for the homogeneous material with permeability $k = 10^{-6}$, a value of 2.06 L/h was obtained, which was much higher than the average value of 0.23 L/h. This resulted in the case of

heterogeneous material. It should be noted that the maximum value for the 10,000 Montecarlo calculation cases was 3.51 L/h, resulting in a low probability (19/10,000 = 0.19%) of obtaining values higher than the homogeneous calculation value of 2.06 L/h.

- In addition, since there is a great dispersion, there is the possibility of finding great variability of permeability between lifts, thus being able to raise the gradient and cause internal erosion problems. In this case, the average value stood at 3.55, which is below the risk limits. It should be noted that maximum values close to 10 were reached with an occurrence probability of (1134/10,000 = 11.34%) of values greater than 5. Therefore, there is a moderately low risk that the core potentially has internal erosion problems. Moreover, the situation did not degrade the construction of core dams with homogeneous material, where in any case the transition to the shoulders must be made using filters that cancel any risk of erosion.

The probabilistic calculation can be extended with an adequate distribution law of adjustment of results. For these purposes and for possible safety and reliability studies, an adjustment of the distribution functions of the maximum seepage flows and gradients obtained in the 10,000 calculations that were performed are shown in Figure 22.

The adjustments were made by means of sigmoid curves, using the expression below:

$$N = \frac{a}{1 + e^{b+cX}}$$

where $N$ is the number of calculations, $X$ is the variable that is calculated in each case (the seepage flow or the maximum gradient), and $a$, $b$, $c$ are adjustment parameters.

According to the adjustments made, the seepage flow was adjusted by means of the parameters: $a = 10,000$; $b = -5.38$; $c = 26.74$.

For the maximum gradient, the sigmoid adjustment function was defined by the parameters: $a = 10,000$; $b = -5.32$; $c = 1.52$.

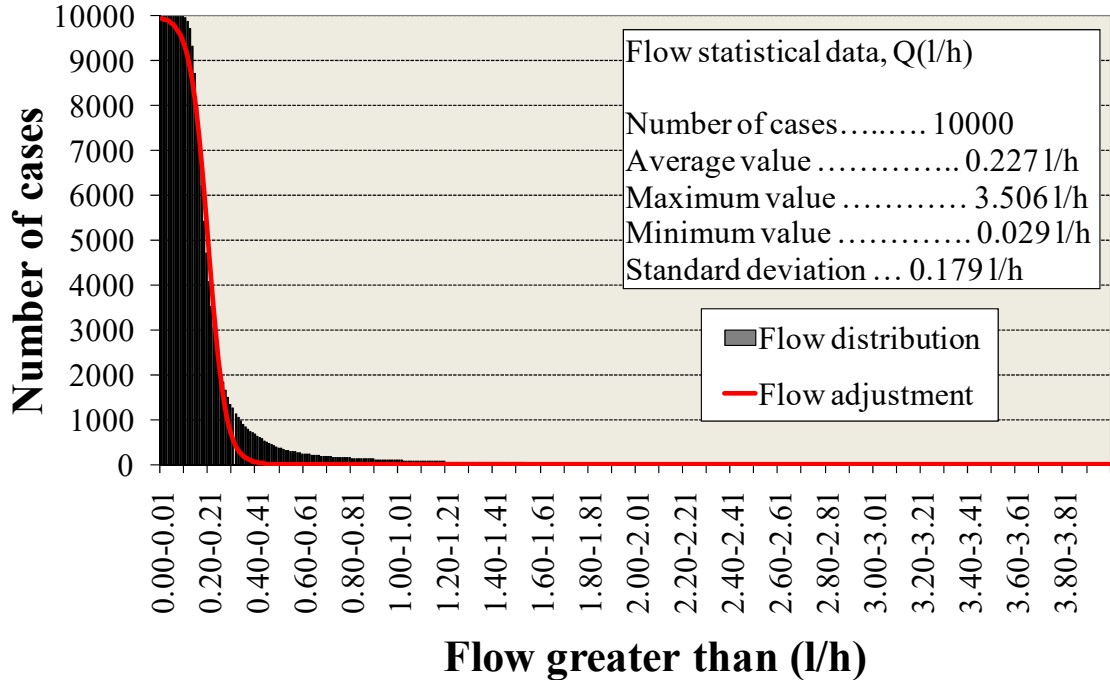

**Figure 22.** *Cont.*

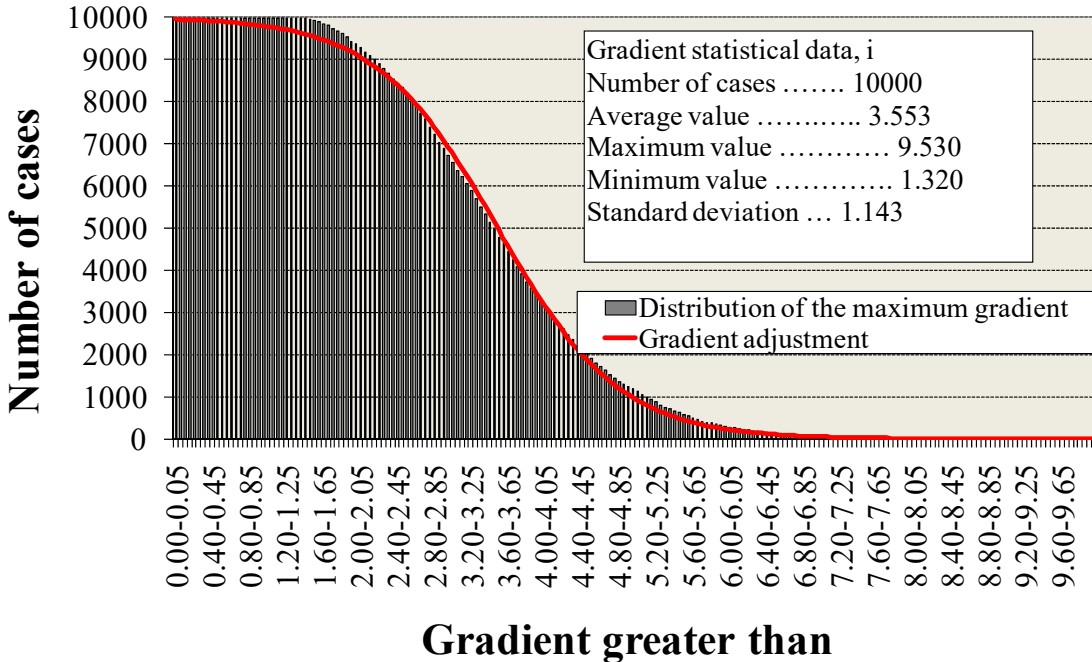

**Figure 22.** Distribution functions adjustments of the maximum seepage flows and gradients obtained for heterogeneous nucleus.

Analogous studies with different mean permeability values and the distribution range of permeabilities could provide evidence to evaluate the probability of failure risk when using this type of heterogeneous materials in the construction of embankment dam cores. For this purpose, a criterion of internal erosion (analysis of maximum gradients) and maximum expected seepage flow rates should be established by comparing the threshold values set with the results obtained from the different calculations.

## 5. Conclusions

To control the seepage, geotechnical recommendations prescribe a high proportion of fines and a high homogeneity of geomechanical characteristics for the material used in the dam core. However, on many occasions, it is not possible to find material of this nature at the dam site or in nearby areas. Unfortunately, using and transporting soil with good geotechnical characteristics to construct the core in a faraway location is economically and environmentally unsustainable. Therefore, it is necessary to minimize the amount of material obtained far from the dam location. To this end, the possibility of using the less suitable material at the dam site as part of the core can be studied with a different core dam design.

In the present research an optimized core dam design was proposed by analyzing two situations that can usually occur: (1) the possibility of using various soil with a marked difference in grain size as the core of the dam, each with homogeneous geotechnical properties; (2) the use of a soil of great heterogeneity in its geotechnical properties, so that it presents a great dispersion of permeability.

In the first case, the sensitivity study of seepage flows for different designs of dam cores, could serve as a reference to establish the requirements for minimal seepage flow rates during the operation of the dam (optimized core dam design). Different designs could take into account the geometry of the core, the height of water in the reservoir, or the permeabilities of the materials. The following conclusions regarding the adequate zoning of the core can be established:

- The maximum seepage flow rates are progressively smaller from case 0 (a single material with horizontal permeability $k_1$), case I (two materials set horizontally with horizontal permeability $k_1$ and $k_2$), case II (three materials set horizontally with horizontal permeability $k_1$, $k_2$, and $k_3$),

and case III (three materials set vertically with horizontal permeability $k_1$, $k_2$, and $k_3$), with this last zoning being the most effective in decreasing the seepage flow, where for all cases: $k_1 > k_2 > k_3$.

- Attention should also be drawn to the fact that the volume of all the materials in case III is approximately the same, thus minimizing the environmental and the economic impact. Whereas in case II and case I, a higher proportion of the more impermeable material is needed to optimize the seepage flow.

- For each case, different sub-cases were also studied by varying the materials in the dam core. In the most effective case, case III, with water heights under 90% of the height of the dam core, the most optimized zoning (minimal seepage flow rates) was obtained for subcases III-A and III-C corresponding to the most impermeable soil situated downstream. However, for the maximum water height, the seepage flow increased exponentially in these two subcases, and zoning III-B and III-E for maximum water height was more optimized, which corresponds to the intermediate permeability material located downstream.

In the second situation (2), seepage flow rates and maximum hydraulics gradients in the core are essential when heterogeneous materials are used to construct the impervious element of dams, especially when using poorly known materials that present uncertainties in their geotechnical characterization, with very extended granulometries and wide ranges of variation of their permeabilities. In this case, the conditions that make these materials possible to use are studied by means of a Montecarlo analysis that, maintaining the global heterogeneity, can study the unlimited dispositions of lifts of different permeability in the core. The main conclusions obtained in this study are as follows:

- Both mean values of the seepage flows and mean values of the maximum hydraulics gradients had a tendency to stabilize, as the number of results of the available calculations increased.

- The seepage flow rates were limited to sufficiently low values when the average value of the permeability was $10^{-6}$, despite the large dispersion of the permeability of the material. Comparing the result for the homogeneous material with permeability $k = 10^{-6}$, a value of 2.06 L/h was obtained, which was much higher than the average value of 0.23 L/h in the case of heterogeneous material.

- It was possible to verify how the values of the highest maximum hydraulic gradients were not necessarily associated with the values of the highest seepage flow rates. High gradients were obtained for situations in which the seepage flow rates were moderate and low.

- The highest maximum gradients were observed in the lowest lifts of the dam core (near the foundation, height below 1–2 m). In other words, the probability of obtaining the maximum gradient increased when the core height was low.

- Given the wide dispersion, there was the possibility of finding great variability of permeability between lifts, thus being able to raise the gradient and cause internal erosion problems. In the calculations, the average value of the maximum hydraulic gradient was around 3.55, obtaining values below the risk limits and the situation does not worsen for dams with core of homogeneous material, where in any case must be made filters that minimize any risk of erosion.

- The probabilistic calculation can be extended with an adequate distribution law of adjustment of the results. For these purposes, and for possible safety and reliability studies, an adjustment was made to the distribution functions of the maximum flows and gradients obtained in the 10,000 calculations.

**Author Contributions:** Conceptualization, J.S.-M., R.G.; methodology, J.S.-M., R.G.; validation, J.S.-M., R.G., C.A.; writing—original draft preparation, J.S.-M., R.G.; writing—review and editing, I.M.-P.; supervision, L.K., O.S. All authors have read and agreed to the published version of the manuscript.

**Funding:** This research was funded by ARPO, Empresa Constructora S.A.

**Conflicts of Interest:** The authors declare no conflict of interest.

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
