# Peer review of "Optimized Design of Earth Dams: Analysis of Zoning and Heterogeneous Material in Its Core"

_sustainability, doi:10.3390/su12166667_

Round 1
Reviewer 1 Report
This manuscript presents the optimization of dam design from the results of many simulation cases with different conditions of hydraulic conductivity. The manuscript is well written and well organized, but have some major issues as described below:
Major issues: Although the numerical model used in this work is relatively simple and based on the Laplace equation, the section for the validation of numerical model is missing in this manuscript. The authors may present model results versus observed data in the field (can be data from any other’s work) before performing the parametric study.
Minor issues: Please write Equation (1) in word by equation function rather than copy and paste as figure. Please do this for the rest of equations throughout the manuscript.
Suggestion: It would be great if authors can add the figure describing dispersion of hydraulic conductivity versus the potential of internal erosion so that future readers will have some insight on the impact of heterogeneity of soils on internal erosion of the dam. The paper discusses the results based on the well known fact that high hydraulic gradient causes internal erosion, but did not present quantitative description of internal erosion. This suggestion would improve the novelty and contribution of this work.
Author Response
This manuscript presents the optimization of dam design from the results of many simulation cases with different conditions of hydraulic conductivity. The manuscript is well written and well organized, but have some major issues as described below:
Point 1: Major issues: Although the numerical model used in this work is relatively simple and based on the Laplace equation, the section for the validation of numerical model is missing in this manuscript. The authors may present model results versus observed data in the field (can be data from any other’s work) before performing the parametric study.
Response 1: Ok, the reviewer is right. Thank you very much for helping us to improve the paper.
A new section 2.2.3 of the model validation has been incorporated, where the flows measured in the field at Guadalcacin dam (Cadiz, Spain) are compared with the results of the finite difference numerical model described previously and used in the paper.
The comparison shows a reasonable predictive capacity that allows the validation of the numerical model.
Point 2: Minor issues: Please write Equation (1) in word by equation function rather than copy and paste as figure. Please do this for the rest of equations throughout the manuscript.
Response 2: Ok, the reviewer is right. All the equations are modified and written directly into word.
Point 3: Suggestion: It would be great if authors can add the figure describing dispersion of hydraulic conductivity versus the potential of internal erosion so that future readers will have some insight on the impact of heterogeneity of soils on internal erosion of the dam. The paper discusses the results based on the well known fact that high hydraulic gradient causes internal erosion, but did not present quantitative description of internal erosion. This suggestion would improve the novelty and contribution of this work.
Response 3: Indeed, the reviewer indicates the idea that we are currently developing as a continuation of this paper.
To represent the hydraulic gradient versus the potential of internal erosion we need to previously define a criterion of internal erosion by comparing the threshold values set with the results obtained from the different calculations. In the literature there are various criteria for internal erosion that obviously depend on said hydraulic gradient but also on other variables (such as granulometry, compaction, effective stress, etc).
We are currently working in this line of research using rigorous criteria (from research in scientific journals) that define internal erosion in order to offer internal erosion risk diagrams based on heterogeneity. However, that requires much more analysis and extends beyond the scope of this first paper.
We sincerely appreciate the reviewer's suggestion, since it encourages us to continue along this line of research in which we are already working for a recent next publication.

Reviewer 2 Report
The topic is interesting and important on “Optimized Design of Earth Dams Using Heterogeneous Material in Its Core”. The purpose of this manuscript is important numerical results are interesting. However, unfortunately, the manuscript is difficult to understand due to insufficient presentation. The manuscript cannot be recommended for publication in its current state. I have carefully reviewed the paper. I think this paper has not an enough quality to be accepted by “Sustaninability”. It has important technical and writing lacking. The organization of the paper is poor. The material and methodology details are insufficient.
Here below please find a series of comments that the authors may want to consider to improve the clarity of their manuscript
- Title of paper should change with reflects the paper main research.
- Abstract: The abstract is weak and need to be enhanced. Results are not outlined in the abstract. Authors must include briefly their findings.
- Page 3, Figure 1, the author’s show that “the water level up to top level of dam (h = H)”. Authors should explain scientifically, why water level up to top at reality?
- Page 3, the author’s stat that “The soil was considered incompressible, therefore, its compactness did not vary throughout the filtration process”. Authors should explain scientifically, why your hypothesis soil is incompressible? This hypotheses is not correct,
Soil is highly compressible.
- There is some confusion for reader in heterogeneous of materials. Authors should explain in details geotechnical parameter which consider heterogeneous.
- Main purpose of the contribution?
What I am missing is a clear message – if authors have 1 minute to tell someone why this work is exciting and why they should read it – what would you say. Right now it is not clear. For that it is actually really important to write the abstract – What is it that authors think is the main advanced with this work?
- We use normally the introduction to bring the reader up to speed what is known in the subject area and where the gaps are. Authors need to work on the introduction as this does not come across: What is known in terms of filters, drain, geotechnical materials parameter, erosion at different hydrostatic pressure, numerical models and their prediction and mechanics behind deformation or crack/erosion, and where the gaps are.
- Section 3.3 material permeability, authors consider k = 10-3- 10-7 cm/s, which not clear why this values choose this value for analysis. Authors should explain scientifically
- All figures are not clear and good quality. Authors should improve the quality of figures.
- All the results and discussion are rather general, should be explained clearly and scientifically, water pressure, permeability of material, seepage path, dam core geometry etc. practical communities can understand easily and implementation will be easy in practical field.
- Authors should also review the conclusions thoroughly.
Author Response
The topic is interesting and important on “Optimized Design of Earth Dams Using Heterogeneous Material in Its Core”. The purpose of this manuscript is important numerical results are interesting. However, unfortunately, the manuscript is difficult to understand due to insufficient presentation. The manuscript cannot be recommended for publication in its current state. I have carefully reviewed the paper. I think this paper has not an enough quality to be accepted by “Sustaninability”. It has important technical and writing lacking. The organization of the paper is poor. The material and methodology details are insufficient.
Here below please find a series of comments that the authors may want to consider to improve the clarity of their manuscript
Point 1: Title of paper should change with reflects the paper main research.
Response 1: Indeed, the reviewer is right. Thank you very much.
The original title did not include the zoning analysis. Title has been changed to: "Optimized Design of Earth Dams: Analysis of Zoning and Heterogeneous Material in its Core”.
Point 2: Abstract: The abstract is weak and need to be enhanced. Results are not outlined in the abstract. Authors must include briefly their findings.
Response 2: Ok, the reviewer is right. The abstract has been rewritten and the main conclusions and results obtained in the paper have been included.
Point 3: Page 3, Figure 1, the author’s show that “the water level up to top level of dam (h = H)”. Authors should explain scientifically, why water level up to top at reality?
Response 3: Ok. As the reviewer indicates, Figure 1 can be confusing, so a comment has been added regarding the height of the water.
In this research, a zoning analysis of the dam core has been carried out, varying different problem conditions such as slope inclination, permeability of materials and water height, among others.
Figure 1 is only a diagram of a situation studied, since to account for different situations of the reservoir and to be able to see the sensitivity of the filtration flow results with the increase in water, five possible water heights were modeled in the different calculations of the study: h = H, h = 0.9H, h = 0.8H, h = 0.7H and h = 0.6H.
The numerical modeling carried out includes only the core of the dam, which is where the filtration occurs; although evidently considering the shoulders of the dam, top width of the dam can be located higher. It must be said that in the parametric analysis the study has been limited to a certain top width, so that the study can be extended using more widths. Thus, for example, for dam heights of 100 m, a top width of the core equal to 10 m has been considered, if for example it were 8 m then the top of the core of the dam would be at 103 meters and we would be studying a case with 3 meters of freeboard (being the result of the filtration is the same as that studied with 10 m of top width top and without freeguard).
A comment is added in section 2.1 to clarify it.
Point 4: Page 3, the author’s stat that “The soil was considered incompressible, therefore, its compactness did not vary throughout the filtration process”. Authors should explain scientifically, why your hypothesis soil is incompressible? This hypotheses is not correct,
Soil is highly compressible.
Response 4: Ok, as the reviewer indicates, the hypothesis is clarified and it is stated in a more scientifically precise way.
Indeed, the soil that makes up the body of the dam is compressible and the calculation of deformations with an adequate constitutive model and soil parameters would allow the movements to be obtained.
What is intended to indicate in this hypothesis is that the soil compressibility does not influence the calculation of the filtration. In other words, we carry out, as usual in this type of study, a filtration calculation decoupled from the stress-strain analysis of the soil.
Thus, in section 2.1 the previous statement is changed to the following: "The filtration calculation is considered decoupled with the stress-strain analysis. Therefore, its compactness did not vary throughout the filtration process."
Point 5: There is some confusion for reader in heterogeneous of materials. Authors should explain in details geotechnical parameter which consider heterogeneous.
Response 5: Ok. As the reviewer indicates, the concept of heterogeneous material indicated in this research is clarified and the material used in the calculations of the paper is rigorously defined and with geotechnical parameters.
In general, a heterogeneous material shows highly variable properties at source and its characterization presents significant uncertainties: these materials have a very extensive granulometry, with the presence of particles of very different sizes, with permeabilities that vary over a wide range.
In our case, when we refer to heterogeneous material for filtration analysis, we define the material that presents a great dispersion of permeability.
Specifically in section 4.3, when defining the permeability of the material, we indicate the geotechnical characteristics considered for the study of filtration of the heterogeneous material. In our case study, we set an average permeability value of 10-6 cm/s and a variation of the permeability of the material between 6.15 • 10-14 cm/s of minimum value and 5 • 10-3 cm/s of maximum value.
This explanation is incorporated in section 4.3.
Point 6: Main purpose of the contribution?
What I am missing is a clear message – if authors have 1 minute to tell someone why this work is exciting and why they should read it – what would you say. Right now it is not clear. For that it is actually really important to write the abstract – What is it that authors think is the main advanced with this work?
Response 6: We sincerely appreciate the comment to allow us to better express the objective and progress that is achieved with this research.
The main advance of this research is to demonstrate that it is possible to use as an impermeable material in earth dam cores, soils that are considered unsuitable according to the classic recommendations and guidelines. In this way, in locations where the ideal impermeable material is scarce, it can be economized in some cases and make the construction of dams viable in others using the material from the dam site or in nearby areas.
We include this explanation in the abstract and the sections of the manuscript where we indicate the objective of the paper.
Point 7: We use normally the introduction to bring the reader up to speed what is known in the subject area and where the gaps are. Authors need to work on the introduction as this does not come across: What is known in terms of filters, drain, geotechnical materials parameter, erosion at different hydrostatic pressure, numerical models and their prediction and mechanics behind deformation or crack/erosion, and where the gaps are.
Response 7: Ok. Following the indications of the reviewer, the introduction is completed with some existing gaps in the scope of the paper.
Point 8: Section 3.3 material permeability, authors consider k = 10-3- 10-7 cm/s, which not clear why this values choose this value for analysis. Authors should explain scientifically.
Response 8: Ok, the reviewer is right. An explanation is included in section 3.3.
The material considered desirable as a dam core must have a maximum permeability of 10-5 cm/s, since above these values the material is considered semi-pervious (Hatanaka et al. [27] and Murray [28]). Therefore, with this mean reference value, other materials of lower and higher permeability are considered than combined according to different zoning may together have a filtration flow equal to or less than that corresponding to a single material of 10-5 cm/s.
Point 9: All figures are not clear and good quality. Authors should improve the quality of figures.
Response 9: Ok. The quality of all the figures is improved as indicated by the reviewer.
Point 10: All the results and discussion are rather general, should be explained clearly and scientifically, water pressure, permeability of material, seepage path, dam core geometry etc. practical communities can understand easily and implementation will be easy in practical field.
Response 10: Ok. The reviewer's indications are followed for an easier understanding and interpretation of the results obtained and graphs presented.
Thus, in order to make the use of the design graphics more practical, explanations are incorporated that facilitate the use and interpretation of all the figures presented in the discussion of the results included in section 3.6.
Point 11: Authors should also review the conclusions thoroughly.
Response 11: Ok. The reviewer is right. There is some confusion in the statements of the conclusions. Thank you very much.
The conclusions have been rewritten, clearly differentiating what was obtained in the core zoning calculations and the conclusions for the hydraulic gradient and filtration flow in the case of heterogeneous material.
Some conclusions have also been clarified, indicating more clearly the zoning compared according to the permeability values. In addition, the optimized design is clearly indicated as the one corresponding to lower values of filtration flow in each used zoning.
Results have also been clarified in the conclusions, differentiating mean and maximum values in flow rates and gradients.

Round 2
Reviewer 1 Report
The authors well addressed my comments in the first round review and the paper is now better organized and have necessary information for future readers.